

# Size-dependent response of foraminiferal calcification to seawater carbonate chemistry

Michael J. Henehan[1], David Evans[1,2], Madison Shankle[1], Janet Burke[1], Gavin L. Foster[3], Joseph Durrant[3], Eleni Anagnostou[3], Thomas B. Chalk[3], Joseph A. Stewart[3,4], Claudia H. S. Alt[3,5], and Pincelli M. Hull[1]

[1]Department of Geology and Geophysics, Yale University, 210 Whitney Avenue, New Haven CT-06511, USA
[2]Department of Earth Sciences, University of St. Andrews, Irvine Building, North Street, St. Andrews, Fife, KY16 9AL, UK
[3]Ocean and Earth Science, University of Southampton, National Oceanography Centre Southampton, Southampton, SO14 3ZH, UK
[4]National Institute of Standards and Technology, Hollings Marine Laboratory, 331 Ft. Johnson Road Charleston, SC-29412, USA
[5]Department of Biology, College of Charleston, Charleston SC-29424, USA

*Correspondence to:* Michael Henehan & David Evans (michael.henehan@yale.edu; de32@st-andrews.ac.uk)

**Abstract.**

The response of the marine carbon cycle to changes in atmospheric $CO_2$ concentrations will be determined, in part, by the relative response of calcifying and non-calcifying organisms to global change. Planktonic foraminifera are responsible for a quarter or more of global carbonate production, therefore understanding the sensitivity of calcification in these organisms

to environmental change is critical. Despite this, there remains little consensus as to whether, or to what extent, chemical and physical factors affect foraminiferal calcification. To address this, we directly test the effect of multiple controls on calcification in culture experiments and core-top measurements of *Globigerinoides ruber*. We find that two factors, body size and the carbonate system, strongly influence calcification intensity in life, but that exposure to corrosive bottom waters can overprint this signal post mortem. Using a simple model for the addition of calcite through ontogeny, we show that variable body

size between and within datasets could complicate studies that examine environmental controls on foraminiferal shell weight. In addition, we suggest that size could ultimately play a role in determining whether calcification will increase or decrease with acidification. Our models highlight that knowledge of the specific morphological and physiological mechanisms driving ontogenetic change in calcification in different species will be critical in predicting the response of foraminiferal calcification to future change in atmospheric $p$CO$_2$.

# 1   Introduction

Calcium carbonate ($CaCO_3$) production and transport to the deep ocean (the so-called 'carbonate pump') is one of the most important sinks of carbon, acting across a range of geological timescales (Sigman et al., 2010; Berner and Raiswell, 1983). In the Cenozoic (0-66 Ma), biogenic $CaCO_3$ production by foraminifera, coccolithophores and coral reef ecosystems comprises the vast majority of marine carbonate production (Ridgwell and Zeebe, 2005). The strength of this carbonate pump can be



altered in three principle ways: 1) by changing the efficiency of inorganic and/or organic carbon export and burial, 2) by changing the absolute or relative abundance of calcifying and non-calcifying taxa, and 3) by changes in the calcification efficiency of marine calcifiers. All three factors are thought to be sensitive to environmental conditions (e.g. Beaufort et al., 2011; Barker and Elderfield, 2002), although the exact nature of this environmental sensitivity remains unclear. Here we use a series of

culturing experiments to specifically address how pH change can influence the extent to which foraminifera calcify their tests (i.e. their 'calcification intensity').

Metrics of calcification in planktonic foraminifera have already been the subject of much scientific attention because of both the importance of foraminifera to the global carbonate burial flux (32-80% of the total deep marine calcite budget Schiebel,

2002) and the potential of these metrics to act as proxies for changes in marine carbonate chemistry. In this manuscript we will refer to several types of related (but distinct) metrics that have been used to describe calcification in foraminifera, and so for clarity these are summarised in Table 1. Foraminiferal size-normalised weight (SNW) has variously been suggested as a tracer of both changes in deep sea carbonate dissolution (e.g. Lohmann, 1995; Broecker and Clark, 2001) and as a proxy for changes in the surface ocean carbonate system (e.g. Barker and Elderfield, 2002; Bijma et al., 2002; Moy et al., 2009; Marshall et al.,

2013). As a proxy of surface conditions, it is thought that the SNW of foraminiferal tests reflects the carbonate saturation state of the shallow waters inhabited during their lives. Conversely, studies of sea-floor dissolution relate SNW to carbonate diagenesis after death in undersaturated bottom waters. In both cases, studies implicitly assume that either the conditions during life, or conversely, after death, have a relatively minor effect on SNW, or can be accounted for by other means (as discussed by Bijma et al., 2002; Barker and Elderfield, 2002). Whilst both culture and field studies support a surface carbonate system

control on foraminiferal calcification (Bijma et al., 2002; Barker and Elderfield, 2002; Russell et al., 2004; Lombard et al., 2010; Marshall et al., 2013), others have observed secondary environmental controls on SNW such as nutrient availability or temperature (e.g. Aldridge et al., 2012; Weinkauf et al., 2016). Furthermore, other studies observe an inverse response of SNW to carbonate system change in some species (Beer et al., 2010b; Foster et al., 2013) - that is, a greater test thickness at lower carbonate saturation state, pH and/or [$CO_3^{2-}$].

It is possible that at least some of the discrepancies described above may stem from methodological differences, since foraminiferal SNW has been quantified in a number of different ways (see Beer et al., 2010a; Aldridge et al., 2012; Marshall et al., 2013, for further discussion). Many early studies used sieve-based weight measurements, where SNW is calculated as the measured mass of pooled individuals within a set sieve-size fraction divided by the number of individual tests (e.g. Lohmann,

1995; Broecker and Clark, 2001). However, shell size can vary within a studied sieve range (Beer et al., 2010a). Many later studies circumvented this issue by directly measuring the major axis length (Barker and Elderfield, 2002; Aldridge et al., 2012; Beer et al., 2010a) or cross-sectional area (Beer et al., 2010b; Marshall et al., 2013, 2015; Osborne et al., 2016; Weinkauf et al., 2013, 2016) of each individual within a sampled population. However, as discussed by Weinkauf et al. (2016), an assumption common to most shell-weight studies is that SNW metrics themselves do not vary as a function of size – which is unlikely

to be true. It is thought that every time a new chamber is added, foraminifera thicken the calcite of their previous chambers





(e.g. Be and Lott, 1964; Erez, 2003), so that a given chamber will appear increasingly heavily calcified over the course of an individual foraminifera's life. It is therefore possible that size may contribute to variability in calcification responses recorded between and within studies.

Attempts to reconcile these varied experimental and observational SNW data with each other and with foraminiferal biomineralisation models (de Nooijer et al., 2014, and references within) are still broadly lacking. To address this shortfall, we show here how a simple calcification model can be used to provide a theoretical framework for SNW metrics. We then present new observations from core-top measurements and culture experiments in the shallow-dwelling symbiont-bearing species *Globigerinoides ruber* (Henehan et al., 2013; Evans et al., 2016a, b), in the light of this new modelled framework. We discuss the

implications of our modelling and empirical observations both for explaining the often conflicting results in previous studies, and for predicting the response of planktonic foraminifera to future global change.

## 2   Methods

### 2.1   Culturing

Data from *Globigerinoides ruber* (white) used in this study are collated from numerous experiments across a range of tem-

perature, pH and major ion seawater chemistries, cultured at the Interuniversity Institute of Eilat between January 2010 and November 2013. A detailed description of culturing methods is provided elsewhere (Henehan et al., 2013; Evans et al., 2016a, b). Briefly, for all experiments, foraminifera were towed from the Gulf of Aqaba (Eilat) (depth <20m, temperature 22-24 °C, salinity ∼ 40.4 psu), and cultured in individual 120 mL airtight flasks within temperature-controlled water baths. Illumination was provided by a metal halide lamp (420 W) at levels of ∼200 mmol photons $m^{-2}s^{-1}$ (13 h light:11 h dark), equivalent to

irradiance at 15-20 m depth in the open waters of the northern Gulf of Aqaba (Shaked and Genin, 2006). Every 1-2 days, individuals were transferred to a petri dish, measured using an optical micrometer under a Zeiss inverted light microscope, and fed a juvenile brine shrimp. After gametogenesis, foraminifera were rinsed in deionised water, dried and stored for subsequent analysis.

Here we consider changes in calcification intensity in foraminifera that were originally cultured to calibrate two different geochemical proxy systems: boron isotopes and Mg/Ca (Henehan et al., 2013; Evans et al., 2016a, b). The experimental design varies slightly between these two cases, so we discuss any such differences where they arise. In all experiments, culture solution pH was determined using an electrode calibrated against NBS buffers. The same electrode was also used to measure the pH of a range of prepared seawater solutions which were subsequently analysed for Dissolved Inorganic Carbon (DIC)

concentrations and Total Alkalinity (TAlk), allowing us to cross-calibrate our electrode-derived pH values against calculated pH on the total scale using CO2sys.m (van Heuven et al., 2011), and the constants of Dickson (1990), Lueker et al. (2000), and Lee et al. (2010). Because pH control for boron isotope experiments was paramount, in these experiments each individual flask pH was measured every 2-3 days, and flasks that experienced pH drift had culture seawater solution replaced from a stock





solution stored throughout the experiment in airtight bottles without headspace. Uncertainty on pH is therefore calculated as two standard errors on the mean of each culture flask pHs during cultures. For those experiments intended to test Mg incorporation (Evans et al., 2016a), pH monitoring during culture was less frequent, and this is reflected in a more conservative approximation of pH uncertainty in these cultures. In total, we collate calcification intensity data from 11 separate culture

experiments, with temperatures ranging from 22.8 to 27.8 °C, seawater Mg/Ca ratios ranging from 2.17 to 6.25, and pH (total scale) ranging from 7.54 to 8.20. Some estimates of future anthropogenic ocean acidification suggest a pH drop of 0.5 units by 2100 (Caldeira and Wickett, 2003), and so this wide pH range allows us to investigate the possible changes in calcification over the next century.

## 2.2 Deriving Weights from Cultures

Dried cultured foraminifera were imaged and major and perpendicular axes measured using Macnification software (Orbicule Inc., v2.0). They were then weighed individually on microbalances at Royal Holloway University of London (Exps. DE3 & DE4), Yale University (Exp. MH2), and University of Bristol (Exp. MH1). Mean uncertainty assessed by 2 standard deviations of triplicate measurements of individual foraminifera was $< \pm 1$ $\mu$g. While these foraminifera were not ashed to remove any remnant organic matter prior to weighing, previous comparisons of non-ashed sample weights with mass of $CaCO_3$ determined

by ICPMS showed no significant difference (Henehan et al., 2013).

Since it is unfeasible to take direct shell weight measurements from live pre-culture foraminifera without harming the organism, we estimate pre-culture shell mass from test size (major and perpendicular axes measured via ocular micrometer) at the beginning of culture, using a locally-defined, single-species size-weight relationship, as in previous studies (e.g. Kisakürek

et al., 2008; Lombard et al., 2010; Henehan et al., 2013). Here we expand on our previous size-weight calibration dataset, combining a total of 205 measurements from individuals of *G. ruber* (ranging in size from 141 - 517 $\mu$m) towed from the Gulf of Aqaba (Eilat). The equation of this line ($R^2 = 0.61$) is given in Equation 1 below.

$$\text{shell mass (in } \mu\text{g)} = 217 \times (\text{product of axes, in mm}^2)^{1.43} \tag{1}$$

## 2.3 Defining a calcification metric for cultured foraminifera

Existing metrics for calcification (as discussed above in Sect. 1), cannot be applied directly to laboratory cultures of planktonic species, as many chambers are precipitated prior to collection. Previous studies have therefore used mean test weight of cultured foraminifera from a given size range (e.g. Bijma et al., 2002), or made corrections for pre-culture mass and time spent in culture (Lombard et al., 2010) to describe how calcification responded to culture conditions. Here, we developed a new metric, calcification intensity (CI; Eq. 2), that accounts for both the size/mass of foraminifera upon collection, as well as the (often

differential) amount of mass added between culture experiments:

$$\text{CI} = \frac{\Delta\text{mass}}{\Delta\text{area}} \tag{2}$$





Where $\Delta$mass is the difference in mass between the start and end of the culture (expressed in $\mu$g) and $\Delta$area is the difference in product of the major and minor axes (in mm$^2$) between the start and end of the culture. We quantify calcification in cultured foraminifera in this simple way because it allows for more complete consideration of mass grown outside of culture, and relies only on pre- and post-culture dimensions and mass that are routinely measured.

## 2.4    Ontogenetic modelling of calcification intensity

Planktonic foraminifera are single-celled eukaryotes with calcium carbonate tests that show distinct morphological changes throughout ontogeny (Schmidt et al., 2013; Brummer et al., 1987). Foraminifera grow by sequentially adding calcium carbonate chambers along a primary coiling axis, and it is thought that they lay down an extra layer of calcite ('secondary calcite') over existing chambers when a new chamber is formed (Be and Lott, 1964; Hemleben et al., 1989). Over ontogeny coiling behaviour often changes, as do the relative size, shape, thickness and perforations (i.e., 'porosity') of the calcium carbonate chambers (e.g. Schmidt et al., 2013). Finally, when foraminifera reproduce and die, many species (although not *G. ruber*; Caron et al., 1990) precipitate a thick final layer of carbonate known as gametogenic calcite. Other species appear to secondarily thicken their tests following precipitation of their final chamber (Spero et al., 2015; Fehrenbacher et al., 2016), although this has not yet been observed in *G. ruber* (C. V. Davis, pers. comm.). Studies of calcification in foraminifera must therefore disentangle the effects of various environmental factors from these known ontogenetic phenomena. In open-ocean studies (e.g. sediment trap, core-top), SNW measurements are commonly taken from within narrow size ranges to minimise the effect of size-dependent variation in calcification (as discussed in Beer et al., 2010a; Weinkauf et al., 2016). However, normalisation for size effectively assumes that there is no ontogenetic variation in calcification or that all individuals come from the same ontogenetic stage, which is unlikely, particularly when comparing results across different studies. In laboratory cultures, there are additional difficulties in measuring CI as it is generally not possible to select a narrow starting size range (given specimen limitation), and the number of chambers added by each individual over the course of culture experiments is highly variable. Therefore, to provide a quantitative framework for exploring the relationship between CI and ontogeny, and to explore how existing calcification metrics may be biased by the use of large or variable size fractions, we developed an ontogenetic model of CI using empirical observations from *G. ruber*.

### 2.4.1    Model parameter constraints

To ground the model, morphological measurements of 39 specimens were taken from a natural *G. ruber* population from a Woods Hole Oceanographic Institution core-top sample (sample KC78) from the equatorial Atlantic (5.267°N by 44.133°W, 3273 m water depth; Sun et al., 2006). Selected specimens range in size from 250-600 $\mu$m (150-250 $\mu$m (n=11), 250-300 $\mu$m (n=11), 300-425 $\mu$m (n=10), and 425-600 $\mu$m (n=7)). Foraminifera were first mounted and imaged using a light microscope and ImageJ (www.imagej.net), so that major and minor axis measurements could be taken from as many chambers as possible, as well as from the whole test. Individual chambers were subsequently broken and removed so that chamber wall cross sections could be imaged and measured. Wall thickness measurements were made away from sutures and chamber apertures, as wall



thickness often varies in these regions of the test (Bé, 1980). All reported values are the mean of three replicate measurements, and are given in Supplementary Table 1.

### 2.4.2  Model description

Building upon the measurements described in Sect. 2.4.1 and other published data (listed in Table 2), we designed a simple

calcification intensity model in Matlab that tracks the cumulative calcium carbonate in an idealised foraminifer throughout its life cycle. For a full description of the model, see Appendix A (note the Matlab code also accompanies this paper). In short, this model simulates the addition of $CaCO_3$ mass with growth, as scaled to numerous morphological and wall-thickness parameters. Morphological parameters used to determine carbonate content include chamber size, chamber shape, and chamber overlap. Wall thickness parameters include the thickness of the initial chamber wall prior to any subsequent thickening (the 'primary

wall'), wall porosity, and the thickness of secondary layers added to preceding chambers. There is no *a priori* knowledge of which (if any) morphological parameter(s) we should expect to vary in response to environmental conditions. To address this, we allowed each parameter to vary randomly within set tolerances and ran the model $10^6$ times. With a set of model runs this large, we could identify which factors were systematically linked to change in CI irrespective of concomitant random changes in others. This is an important advantage of this computationally-inexpensive model over more complex and sophisticated

foraminiferal growth models (e.g. Berger, 1969; Signes et al., 1993). Another critical aspect of our model is that it captures the non-linear behaviour of calcification with ontogenetic growth. Three variant forms of the model (each ran $10^6$ times) are presented here, with wall thickness parameterised either as a function of size (Models 1 and 2) or of chamber number (Model 3). All model variants are built upon open ocean measurements, and are screened against open-ocean populations to ensure they are representative (see Appendix A for a more detailed treatment).

We use the CI-models for two primary ends. First, we explore the effect of ontogeny on CI given size-dependent and chamber-dependent variation in primary wall thickness. Second, we use the ontogenetic predictions to correct for variable chamber addition in culture, given the importance of secondary calcification on shell weight (e.g. Be and Lott, 1964; Erez, 2003). For instance, a foraminifera possessing 8 chambers that is of similar overall test size to a 7-chambered individual will have layered their previous chambers with calcite more times, leading to higher CI at the same body size. Control for variable

chamber addition in culture was achieved by simulating the change in CI with each chamber addition ($\Delta$CI) for a foraminifera growing from a set starting size (a maximum axis of 100 $\mu$m) using all feasible shell growth models.

### 2.5  Core-top Sampling

To supplement modelling and culturing, and to examine the extent to which the physiological controls observed in culture are preserved in fossil assemblages, we also examined SNW in core-top *G. ruber*. Core-top sites and locations are given in the

Supplementary Information, and span a range of bottom water calcite saturation state ($\Omega_{calcite}$) from 0.78 to 4.06. Pre-industrial surface ocean carbonate system conditions for each site were estimated using modern alkalinity relationships (from Lee et al., 2006), air-sea $pCO_2$ disequilibrium (from Takahashi et al., 2009; Gloor et al., 2003) and local hydrography (from Takahashi et al., 2009; Garcia et al., 2010) following Henehan et al. (2013). Deep ocean carbonate chemistry and calcite saturation state




were calculated from DIC and TAlk estimates at depth from Goyet et al. (2000). Carbonate system calculations were carried out as in Section 2.1 above, with pressure corrections from Millero (1995) according to each core site's bathymetry.

For size-normalisation of shell weight measurements, we follow the 'area density' ($\rho_A$) approach of Marshall et al. (2013), as used elsewhere (Weinkauf et al., 2013, 2016; Marshall et al., 2015; Osborne et al., 2016). From each core-top sample, individual specimens (n = 8 - 26) of *G. ruber* were picked from discrete sieve size fractions, imaged (umbilical side up), and weighed, and their cross-sectional or silhouette area determined using Macnification (Orbicule Inc., v2.0). $\rho_A$ was calculated as mass (in $\mu$g) over silhouette area (in mm$^2$). As in Marshall et al. (2013) and Osborne et al. (2016), the mean $\rho_A$ and silhouette area in each core-top sample was then used in multivariate statistical analysis.

## 3   Results

### 3.1   Modelling ontogenetic trends and intrinsic drivers of CI and $\rho_A$

Model parameter combinations within the prescribed tolerances of the size-weight ratios seen in non-cultured populations of *G. ruber* are displayed in Fig. 1, panels a,d,g. Using the validated subsets of each model, we calculated CI as it evolved through the ontogenetic growth of each individual modelled foraminifera (Fig 1, panels b,e,h). All models suggest that CI should increase rapidly with test size up to size of ∼0.05 mm$^2$ (equating to a major axis of approximately 210 $\mu$m). Beyond this size there is considerably more scope for inter-individual (i.e. inter-model run) variability, but most model runs continue to show increasing CI with size throughout the remainder of their ontogeny.

From the perspective of foraminiferal cultures, these models predict a strong dependency of CI on foraminifer size on collection, and also the number of chambers precipitated in culture. For instance, cultures in which foraminifer added three chambers on average are expected to have a higher CI than cultures in which most foraminifera added two chambers. Panels (c), (f) and (i) of Fig. 1 explore this effect of chamber addition on CI. For each model, frequency distributions of modelled change in CI ($\Delta$CI) after two and three chamber additions are shown, relative to a baseline addition of one-chamber and a starting size of 100 $\mu$m maximum axis diameter. The median increase of calculated CI with each chamber addition is 17 $\mu$g mm$^{-2}$, and was largely insensitive to the base model – a similar dependency of chamber addition on CI was observed in models 1-3 (Fig. 1c,f,i). To account for the effect of chamber addition on our culture results, we therefore normalise our culture CI data for the mean number of chambers precipitated in culture (detailed in Sect. 3.2).

In addition to discerning ontogenetic trends in CI, these models also help to constrain which morphological parameters exert the most coherent control on CI. For almost any given model parameter, the full range of CI observed in culture can be produced given the right combination of other morphological parameters. For example, CIs of between 120-250 $\mu$g mm$^{-2}$ can be produced irrespective of the chamber aspect ratio. In other words, whilst a change in chamber aspect ratio can affect CI, any such change can also be offset by a compensatory change in other model parameters. The only exceptions to this in our model are the variation in the coefficients $a$ and $b$ in equation A2 – the parameters that determine the non-linear growth function of





wall thickness. Specifically, very low CIs in adult-sized foraminifera were only observed with $a$ and $b$ at their lowermost and uppermost limit respectively (Fig. A2). In contrast to all other parameters, CI change as forced by the shape of the regression between wall thickness and chamber size/number cannot be compensated for by other precribed ontogenetic parameters.

## 3.2 Culture results

Consistent with results of modelling in Sect. 3.1, CI in our culture experiments varied with test size and chamber addition. A logarithmic regression of individuals' CI versus their size at the beginning of culture experiments (Fig. 2a) yields an $R^2$ of 0.37. Once normalised according to the mean number of chambers added during each culture experiment (see Sect. 3.1 and Fig. 1c above), the coherence of this relationship with size becomes stronger, with an $R^2$ of 0.59 (Fig. 2b). Note that logarithmic regression models were used because modelled CI through ontogeny approximates to a logarithmic relationship across the size range of our cultures (see Fig. 1b).

To investigate the effect of culture conditions on CI, the residuals from the logarithmic regression fit in Fig. 2b (given in Table 3) are compared to environmental parameters (Fig. 3). No significant variation with $Mg/Ca_{sw}$ was observed (Fig. 3), consistent with previous observations that varying seawater [Mg] does change growth rates in foraminifera (Evans et al., 2015, 2016a). Similarly, no effect of temperature was observed (Fig. 3b). A statistically significant correlation with pH was found ($R^2$ = 0.47, $p = 0.01$; Fig. 3c). Other studies will often parameterise carbonate system changes in terms of carbonate ion concentration, $[CO_3^{2-}]$ (e.g. Marshall et al., 2013), or $[DIC]/[H^+]$ (Jokiel, 2011; Bach, 2015). We also tested residual CI against these parameters, using carbonate speciation constants adjusted for changing $Mg/Ca_{sw}$ according to Hain et al. (2015). There was little difference in correlation coefficient in the case of $[DIC]/[H^+]$ ($R^2$ = 0.47 for both variables), but in the case of $[CO_3^{2-}]$ correlations were weaker ($R^2$ = 0.40). We opt to primarily focus on pH in figures and discussions here as a less abstracted parameter, but we cannot rule out $[DIC]/[H^+]$ as being the primary driver, as suggested by Bach (2015).

## 3.3 Core-top results

Core-top results are given in Supplementary Table 2. Multiple linear regression analysis demonstrates that deep ocean carbonate system conditions at the core-top site, test size and morphospecies identity were all statistically significant controls on $\rho_A$ within the set of core-top samples measured here. Together, these three factors could explain 86% of the variance in $\rho_A$ seen in our core-top assemblages (for regression statistics, see Supplementary Table 3). We used the relaimpo R package (Groemping, 2006) to determine relative importance of these factors, and found bottom water pH at the site of deposition to be the strongest determinant of $\rho_A$ (as shown in Fig. 4 a and b, and Supplementary Table 3). Shell size (as parameterised by shell silhouette area; Fig. 4c), and species type within the broader *G. ruber* clade (Fig. 4b,c) were found to be secondary, but nevertheless significant factors. Where both *G. ruber* sensu stricto and sensu lato (a combination of *G. elongatus* and *G. pyramidalis*; Aurahs et al. (2011)) were measured at the same site, *G. ruber* sensu stricto displayed significantly lower values of $\rho_A$ (paired t-test; $p$ = 0.01). Despite the observations from our culture experiments, estimated surface-ocean pH was not found to be a significant control on $\rho_A$ in the core-top samples, and would be excluded from the model by stepwise parameter reduction according to



its Akaike Information Criterion (AIC). We include it here for illustrative purposes only given the focus of our study (Fig. 4), since it has little effect on the overall model fit ($R^2$= 0.86 in both cases), or the relative importance of other factors in the regression. We note also that we tested other deep and surface water carbonate system parameters in lieu of pH ($\Omega$, $\Delta[CO_3^{2-}]$, etc.), but in all cases pH produced in stronger model fits.

## 4  Discussion

### 4.1  Towards understanding the differential response of foraminifera to acidification

Previous studies investigating the effect of acidification on marine calcifiers have so far failed to explain the highly divergent responses both within and between the major groups of calcifiers. For example, coccolithophores were more heavily calcified during geological epochs characterised by higher $CO_2$, and lower pH (e.g. Bolton et al., 2016), even if this result has not always been reproduced in culture (Riebesell et al., 2000; Langer et al., 2009). Additionally, despite observations of decreasing pH and $[CO_3^{2-}]$ negatively impacting calcification in most planktonic and reef-dwelling foraminifera (e.g. de Moel et al., 2009; Marshall et al., 2013; Kuroyanagi et al., 2009) in agreement with results from our cultures (Fig. 2), benthic foraminifera became more heavily calcified or exhibited little response over the Palaeocene-Eocene Thermal Maximum (PETM) and ETM2 (Foster et al., 2013). Examining this finding in the context of our model allows us to investigate the morphological responses to pH change that could produce these patterns, and so begin to form a unifying hypothesis to explain these various apparently contradictory observations. To do this, we must first consider how foraminiferal morphology itself might affect calcification.

Our model predicts that calcification intensity – a metric for how heavily calcified cultured foraminifera are – is dependent on foraminifera size on collection as well as the number of chambers precipitated in culture, as we observe (Figs. 1, 2). But beyond this, varying permutations of model parameters (as laid out in Appendix A) reveals that CI is most strongly dependent on covariation of the coefficients that describe the increase in wall thickness with ontogeny ($a$ and $b$ in Eq. A2). It appears that the overall shape of this regression, rather than either of its constituent coefficients alone, is most important in controlling CI. Furthermore, the size-wall thickness coefficients drive CI in models 1 and 3, but varying maximum wall thickness in isolation has no systematic effect (model 2). These observations lead us to hypothesise that it is not simply that large adult foraminifera lay down less calcite in their walls in response to acidification, but rather that the slope of the regression between wall thickness and chamber diameter is shallower. This mechanism, if correct, would produce two physiological responses: (1) lower pH will result in a thinning of shell walls when the foraminifer is larger, and (2) smaller foraminifera will exhibit ambiguous, or even positive, responses to acidification. These hypothesised responses are represented in Fig. 5.

Although it is perhaps counter-intuitive to envisage such contrasting calcification responses to acidification with size, there is growing empirical evidence to support such a model (see Fig. 5). Culture and field studies indicate that large planktonic and shallow-water benthic foraminifera respond to decreased pH by producing thinner walls (Allison et al., 2010; de Moel et al., 2009, this study). Equally, the more heavily calcified small benthic foraminifera observed at the PETM (Foster et al., 2013)





would also fit with our hypothesis. While biomineralisation pathways differ, recent work favours a similarly positive response of calcification to acidifcation in the much smaller coccolithophores (Bolton et al., 2016; McClelland et al., 2016). Such a hypothesis may have a mechanistic foundation in the physiology of biomineralisation, when one considers two important observations of calcite precipitation in foraminifera. Firstly, based on test oxygen isotope ratios, it seems that foraminifera pre-

cipitate calcite from a species-specific combination of $HCO_3^-$ and $CO_3^{2-}$ (Zeebe, 1999; Uchikawa and Zeebe, 2010). Secondly, small foraminifera do not have internal calcium and/or carbon pools (Nehrke et al., 2013), whereas large foraminifera do (Erez, 2003; ter Kuile et al., 1987, 1989). This is because smaller foraminifera build chambers that require far less material volumetrically, and so they can potentially source the ions required for calcification on the same timescales as chamber precipitation. In this way, lower pH could have less of an effect on their wall thickness, or even favour more heavily calcified chambers, if

associated with a rise in [DIC]. Increased availability of carbon for calcification could promote calcification, decreasing the volume of seawater needing to be cycled to produce a given amount of calcite for chamber formation.

Once planktonic and benthic foraminifera reach a certain size, however, the large amounts of $CaCO_3$ required to build a new chamber necessitates prior storage of carbon internally (Erez, 2003; ter Kuile et al., 1987, 1989). Importantly, the efficiency of

this internal storage mechanism is thought to be related to the ability of the organisms to raise the pH of vacuolised seawater (Bentov et al., 2009; de Nooijer et al., 2009). Therefore, lower seawater pH acts against the efficiency of this carbon concentrating mechanism, meaning that chamber formation is more difficult. This is reflected in the observed response of foraminiferal wall thickness to changes in seawater carbonate chemistry (Figs. 2 and 5).

Size-dependent calcification provides a common mechanism to unify the often variable responses observed in foraminifera to date. At present, however, there are insufficient measurements of changes in calcification and morphology through ontogeny to robustly test this hypothesis, or to re-interpret existing SNW data with any great confidence. Comparative CT scans of foraminifera, including examination of individuals grown under different pH conditions, could provide such a test by constraining ontogenetic variation in calcite (as in Schmidt et al., 2013).

## 4.2  Implications for size-normalised weight (SNW) in foraminifera as a proxy

Published investigations using SNW as an environmental proxy commonly assume that the SNW metrics themselves are independent of size/ontogeny. Our modelling approach shows that there is in fact a strong effect of test size on SNW. Modelled area density ($\rho_A$, a commonly used SNW metric Marshall et al., 2013, 2015; Osborne et al., 2016; Weinkauf et al., 2013, 2016) is shown in Figure 6a as a function of test diameter. Virtually all model runs predict a positive relationship between $\rho_A$ and test

size, although the exact nature of this relationship may vary with model parameterisation. This suggests that at least some of the variability between and within published studies could derive from the widely divergent shell sizes and sieve size ranges used (Fig. 6b). Our models similarly reveal a strong size-dependency in volume-normalised approaches, such as that used by Foster et al. (2013) across the PETM (even if a qualitative assessment of the data of Foster et al. (2013) supports their general





conclusions). All model runs predict that the foraminiferal calcite volume/total volume ratio exhibits a strong dependence on test size (Supplementary Fig. 4).

It has been suggested, given some size-mass (or volume-mass) relationships in planktonic foraminifera appear approximately linear (Weinkauf et al., 2016), that the use of area and volume-normalised weight to estimate SNW is not complicated by spatial

or temporal variations in mean body size. In fact, a linear relationship between mass and area or between mass and volume directly implies that area density *is* size-dependent, given:

$$\text{mass} = k \times \text{mm}^3 \tag{3}$$

Where $k$ is a constant equal to the slope. If volume is approximated from area by raising to the power of $3/2$ (Weinkauf et al., 2016), then:

$$\rho_A = \frac{\text{mass}}{(\text{axis}_1 \text{axis}_2)} \quad \approx \quad \frac{k \times (\text{axis}_1 \text{axis}_2)^{3/2}}{(\text{axis}_1 \text{axis}_2)} \quad \approx \quad k \times (\text{axis}_1 \text{axis}_2)^{1/2} \tag{4}$$

which predicts a linear dependence of $\rho_A$ on body size (given aspect ratio in Table 2), such that:

$$\rho_A \approx k \times \sqrt{\frac{\text{axis}_1{}^2}{1.16}} \quad \approx \quad \text{axis}_1 \times \frac{k}{\sqrt{1.16}} \tag{5}$$

For *G. ruber*, the slope of this relationship between size and $\rho_A$ approximates to $2.6 \times 10^{-3}$. This means that a $100 \ \mu m$ increase in test major axis would lead to an $\rho_A$ increase of $\sim 38 \ \mu g \ mm^{-2}$ due to the change in body size alone. This response is of the

same magnitude as the area density response to carbonate system changes reported in other studies (e.g. Marshall et al., 2013; Osborne et al., 2016). We note also that the higher overall $\rho_A$ observed by Marshall et al. (2013) for *T. sacculifer* in the larger size fraction supports the existence of a positive size effect on $\rho_A$, even if in this study calcification response did not appear to change significantly within large size fractions. Similarly, we also find that cross-sectional area is a significant predictor of $\rho_A$ in our core-top foraminifera (Fig. 4).

We thus provide theoretical, model and empirical evidence for a strong test size control on size-normalised weight metrics. This size-dependence applies equally to other SNW metrics, not just the specific metrics directly discussed here. That said, size-dependence of SNW metrics may be variably manifest, and variably problematic, in real-world datasets. One implication of this finding is that datasets from different studies using different foraminiferal size fractions are not directly comparable.

Given the range of size-fractions used in SNW studies (Fig. 6b), size and ontogenetic stage may explain some of the discrepancies between findings, particularly for those studies using a wide sieve size fraction (e.g. several hundred $\mu m$). The magnitude of increase in SNW (here quantified as $\rho_A$, Fig. 6a) with size is likely strongest at smaller body sizes ($< \sim 350 \ \mu m$) - encompassing size fractions commonly used in both modern calibration and down-core studies (Fig. 6b). With these findings in mind, here we make recommendations of best practice for future studies. Firstly, in death assemblages, where post-gametogenic

foraminifera will likely have added a similar number of chambers within a full life cycle, the influence of ontogeny and body size on SNW can be minimised by using the narrowest possible size fraction of only post-gametogenic individuals, and reporting mean test size. This may of course be challenging over transient climatic events associated with a shift in body size,





such as the PETM – in that case, models of calcification with size and ontogeny are needed. Living assemblages may present other difficulties, as individuals of the same size may have more or fewer chambers, and hence differing amounts of secondary calcification. For pre-gametogenic individuals, then, some estimate of chamber number and overall test size is needed. These additional measurements are necessary because the SNW metrics are inherently dependent on foraminifer size, and because

changes in the carbonate system may have a differential effect through ontogeny (as previously suggested by Aldridge et al., 2012).

Besides the influence of size on SNW, our study highlights other fundamental caveats about the applicability of SNW as a surface-water proxy. Although our culture data supports a primary carbonate system control on calcification, in agreement with

other studies (e.g. de Moel et al., 2009; Marshall et al., 2013; Osborne et al., 2016), our core-top samples demonstrate that this may be overwhelmed by dissolution processes at the site of deposition (see Fig. 4). Indeed, while size and morphospecies are preserved as significant controls on SNW (quantified in this case as $\rho_A$; Fig. 4), any primary signal of surface carbonate chemistry (for a range in pH of 8.09 - 8.21) in our core-top sample set has been entirely lost to dissolution. We therefore urge caution before attributing open-ocean SNW patterns exclusively to either primary or secondary processes. In sites where car-

bonate saturation at the site of deposition is high, it is probably reasonable to attribute SNW changes to surface conditions (as in Barker and Elderfield, 2002; Marshall et al., 2013; Weinkauf et al., 2013; Osborne et al., 2016, , etc.). At depths approaching the lysocline, our data supports the earlier uses of SNW as a proxy for the deep sea carbonate system (e.g. Lohmann, 1995; Broecker and Clark, 2001). In between these end-member scenarios, it may be difficult to untangle competing effects (Bijma et al., 2002), and so combining shallow-water and deep-water cores is advisable, following Barker et al. (2004).

Incidentally, our finding that *G. ruber* sensu lato in core-tops has significantly higher SNW ($\rho_A$) than *G. ruber* sensu stricto (Fig. 4) may help to explain the apparently negative correlation between [$CO_3^{2-}$] and SNW in *G. ruber* observed in tows by Beer et al. (2010b). Beer et al. did not differentiate between species, but it is likely they would have sampled an increasing proportion of higher-SNW sensu lato species (i.e. *G. elongatus* and *G. pyramidalis*) in lower-pH upwelling waters, given these

species preference for colder waters (Steinke et al., 2005). This aside, it is also conceivable that these authors may have sampled individuals of similar size, but with a greater number of smaller chambers, as they approached the centre of upwelling. While we cannot confirm these suggestions, taken together, our observations may help to build a more consistent picture of controls on primary calcification intensity in planktonic foraminifera, including the dominance of sedimentary signals in most deep ocean settings.

**4.3   Significance for Global Biogeochemical Cycling**

Previous investigations into the controls on foraminiferal shell weight have often struggled to determine conclusively which environmental controls, if any, impact foraminiferal calcification, as temperature and [$CO_3^{2-}$] are often correlated in hydro-graphic datasets (Aldridge et al., 2012; Marshall et al., 2013). In our cultures where both temperature and carbonate system parameters were varied, we show that the carbonate system is the most likely the driver of CI in *G. ruber* (Fig. 3). Given the





significance of planktonic foraminiferal tests to the global pelagic $CaCO_3$ budget (Schiebel, 2002), this finding could therefore have important implications for global carbonate alkalinity fluxes, and projections of response of biogeochemical cycling to anthropogenic ocean acidification. Within the pelagic realm, foraminiferal calcification reduces TAlk and DIC in a 2:1 ratio, releasing $CO_2$ and thereby lowering surface ocean pH (e.g. Zeebe and Wolf-Gladrow, 2001). Therefore, it is possible that

reduced alkalinity uptake in surface waters may constitute a weak negative feedback on surface ocean acidification. Scaling this in terms of fluxes, considering changes in other calcifying groups like coccolithophorids, and accounting for body size and population size, is beyond the scope of this study. We do suggest, however, that modelling of the pelagic ecosystem, that includes considerations of the physiological costs to calcification in each group of marine calcifiers and the effects on cumulative fitness relative to other non-calcifying groups, may be critical in addressing current shortfalls in prediction of future

biogeochemical changes (Mora et al., 2013).

## 5   Conclusions

In this study we approach the question of environmental controls on changing foraminiferal calcification intensity from multiple perspectives, incorporating observations from culturing and the open ocean with models of ontogenetic growth. Our models for shell growth suggest that calcification intensity (i.e. mass increase per unit size increase) will change changes as a function

of ontogeny and body size in foraminifera. This finding provides a theoretical framework for interpreting results from culture experiments. After controlling for size and chamber addition, our culture experiments suggest neither temperature, nor seawater Mg/Ca ratios affect calcification intensity, but that acidification significantly reduces calcification in adult-stage foraminifera, supporting previous open-ocean observations (e.g. Barker and Elderfield, 2002; Marshall et al., 2013). Based on our modelling work, we tentatively suggest that carbonate chemistry affects different sized foraminifera differently, with acidification leading

to reduced calcification in larger foraminifera, but conversely exerting little control (or even favouring calcification) in small individuals. While further work is required to verify differential responses in larger and smaller individuals, these model results could help to explain a number of published observations, and serve to stress the importance of considering size and ontogeny when studying foraminiferal SNW. Additionally, our core-top results also highlight the central importance of post-mortem dissolution, followed by body size and species ID, in driving SNW in fossil assemblages. While the effects of lower ocean pH

upon ecosystem-level biogeochemical fluxes are not yet fully constrained, our findings suggest that production of $CaCO_3$ by large planktonic foraminifera in the pelagic realm will likely be reduced by future anthropogenic ocean acidification.

*Author contributions.* Michael Henehan and David Evans cultured, weighed and measured foraminifera, co-drafted the manuscript, and processed data. David Evans devised and constructed foraminiferal calcification models. Michael Henehan directed core-top investigations. Pincelli Hull directed the study, co-designed models and co-drafted the manuscript. Madison Shankle collected and collated shell weight data

and aided in data processing. Janet Burke collected foraminifera shell morphology measurements to ground models. Gavin Foster funded and co-directed foraminiferal culturing expeditions and co-supervised core-top investigations. Eleni Anagnostou, Thomas Chalk, Joseph Stewart and Claudia Alt cultured foraminifera. All co-authors contributed to refining and editing the manuscript.





*Acknowledgements.* We are grateful for the hard work of James Rae and Katy Prentice when culturing experiment 'MH1' presented here. Jonathan Erez is thanked for hosting the culturing work at his lab, and in offering intellectual guidance. Shai Oron, and the staff and students at the IUI in Eilat are thanked for their assistance throughout culturing work. Michal Kucera, Helen Bostock and Bruce Corliss are thanked for the provision of core-top sample materials. We thank the other members of the Hull lab at Yale for constructive input and discussion. MH acknowledges financial support from the Yale Peabody Museum.





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





**Figure 1.** Model output for the three model scenarios described in Sect. 2.4.2. (a) Measured size-mass relationship for *G. ruber* based on towed specimens from the Gulf of Eilat that were not cultured. The residual sum of squares between these data defines which model runs (individual blue lines) are taken as representative of this natural population. (b) CI data of cultured specimens (equation 2), shown in the context of that predicted from the same set of models for the scenario where two chambers were precipitated in culture. All models predict that CI is dependent on body size on collection, which is also our empirical observation. (c) Model CI dependence on the amount of chambers added in culture. Broadly, the more chambers precipitated, the higher the resultant measured CI. Because of this finding, we use the model relationship between CI and the number of chambers added in culture to normalise the CI data shown in panels b.





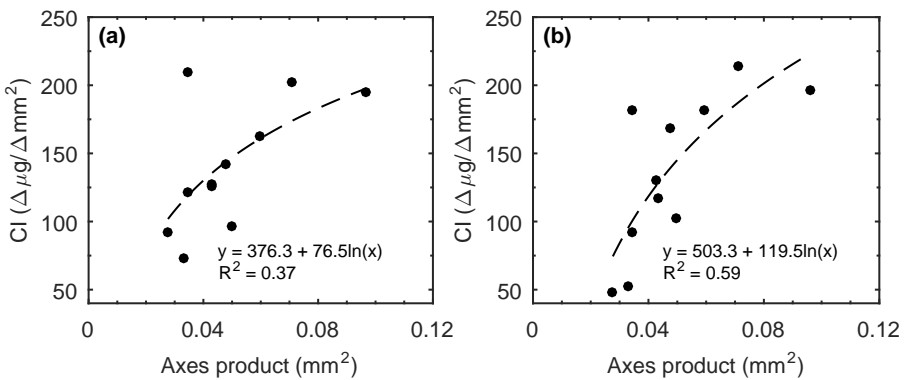

**Figure 2.** (a) Measured CI as a function of the product of major and minor axes at the start of culture for all experiments. (b) CI data can normalised for the number of chambers added in culture, according to the median CI increase of 17 $\Delta\mu$g/$\Delta$mm$^2$ per chamber added from Fig. 1c,f,i. In this case, data were normalised to two chambers added during culture, which was mode for our culture experiments.



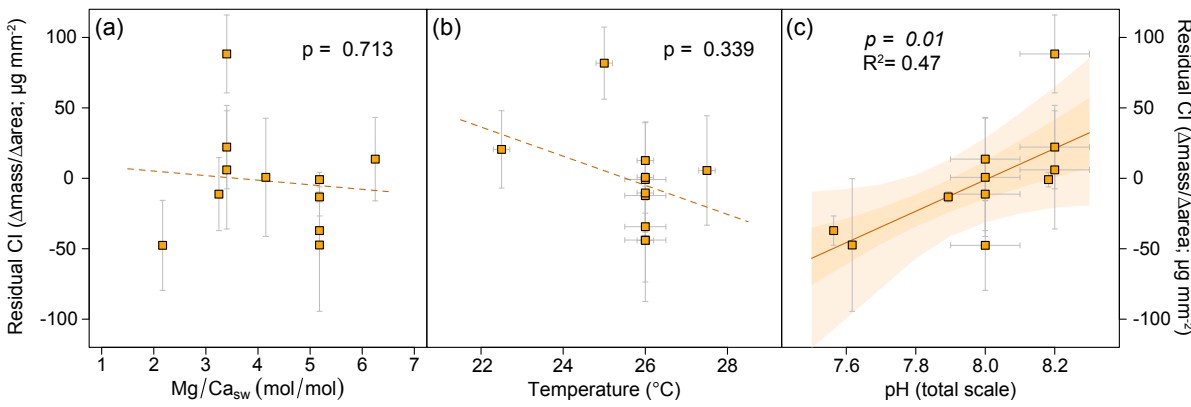

**Figure 3.** Ambient environmental response of chamber-addition corrected residual CI (change in mass/change in area compared to the logarithmic regression through all experiments shown in Fig. 2, see text for justification). (a) $Mg/Ca_{sw}$, (b) temperature, (c) pH. Shaded regions are $1\sigma$ and $2\sigma$ bounds of uncertainty, as calculated via combined wild bootstrap and Monte Carlo analysis, accounting for error in X and Y variables, following Henehan et al. (2016). Dashed lines indicate non-significant relationships, solid regression lines are significant to $\alpha<0.05$. Shaded regions of uncertainty are 1 and 2 sd of 1000 Monte Carlo linear regression models through randomly simulated datasets sampled within the given $2\sigma$ X- and Y-error margins for each sample.





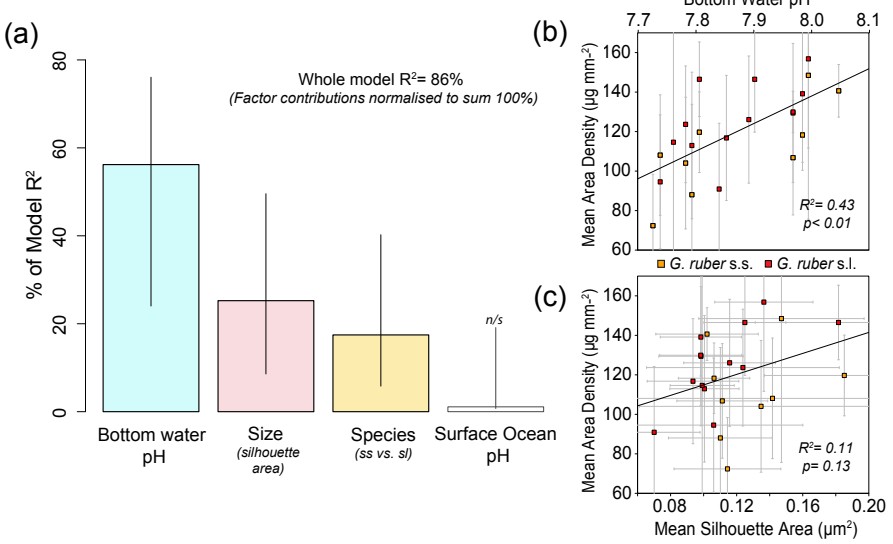

**Figure 4.** Drivers of $\rho_A$ in core-top *G. ruber*. In panel (a), the relative importance of environmental (surface and deep ocean pH) and physiological (size and species) factors in predicing $\rho_A$ are shown. A multiple linear regression model containing bottom water pH (i.e. pH at the site of deposition), test size (i.e. mean silhouette area in the core-top sample) and species (i.e. *G. ruber* sensu stricto vs. sensu lato) can describe 86% of the variance in $\rho_A$, with all predictors significant to $p < 0.01$ except surface ocean pH, which is not a significant contributor to the model but is shown here for illustrative purposes only (see also Supplementary Table 3). Of these variables, bottom water pH is the strongest correlate, as determined by the R package *relaimpo* (Groemping, 2006). Uncertainty bounds on the relative contribution of each variable are determined via bootstrapping. Univariate regression lines of $\rho_A$ vs. bottom-water pH and vs. mean test size are shown in panels (b) and (c) respectively. is shown. Error bars on $\rho_A$ are 2 standard deviations of variation within core-tops (panels b and c). Error bars on mean silhouette area (panel c) are 2 standard deviations of silhouette area within a core-top seive size fraction. In each case, *G. ruber* sensu stricto are plotted in orange, and sensu lato in red. Generally heavier values of $\rho_A$ are observed in *G. ruber* sensu lato (panels b, c), constituting the third most important control on $\rho_A$ within our core-top dataset (panel a). Mutliple linear regression statistics are given in Supplementary Table 3. Residual variation in $\rho_A$ around the relationship with bottom water pH is significantly correlated with test silhouette area ($R^2=0.35$, $p< 0.01$; see Supplementary Fig. 5). Note, other deep ocean carbonate system parameters ($\Omega$, $\Delta[CO_3^{2-}]$) were also trialled in multiple regression models, but the strongest correlation was observed with pH at each core-top site, and so that variable is preferred here.





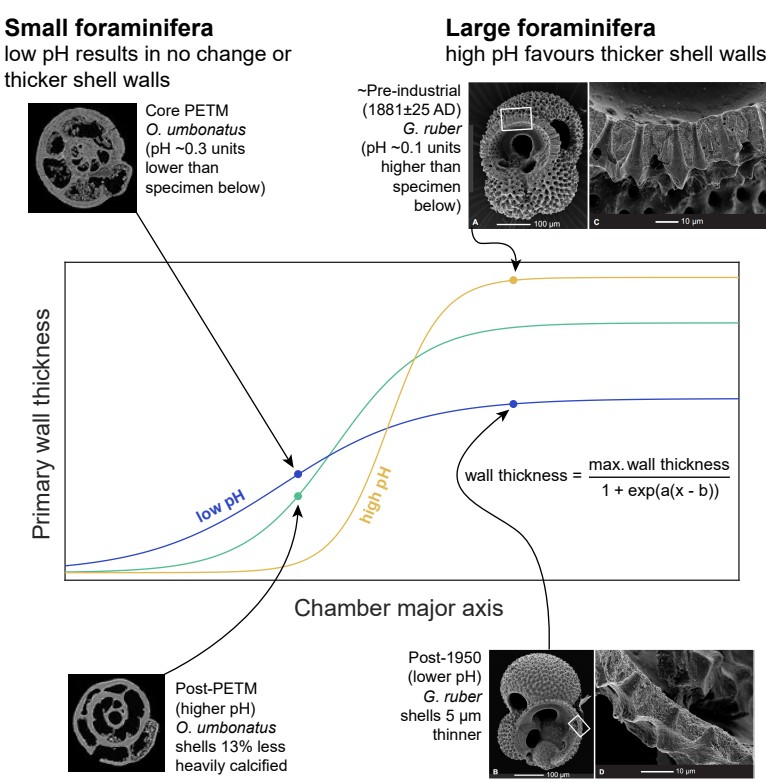

**Figure 5.** Schematic response of the chamber wall thickness-chamber size slope with carbonate chemistry, based on our observations that (1) CI is controlled by pH in culture, and (2) modelled CI responds principally to the slope of this relationship, where shallower slopes are characterised by lower CI. These findings potentially reconcile the differential response of small and large foraminifera to acidification. For example, *G. ruber* in culture (this study) and the Arabian Sea (de Moel et al., 2009) respond negatively to reduced pH. In contrast small benthic species exhibited little or the opposite response to acidification (blue line) over the PETM and ETM2 (Foster et al., 2013) is more enhanced calcification, compared to post-PETM carbonate system rebound (green line). Images of *O. umbonatus* reproduced from Foster et al. (2013) with publishers' consent; images of *G. ruber* from de Moel et al. (2009) reproduced under CC3.0 licence.



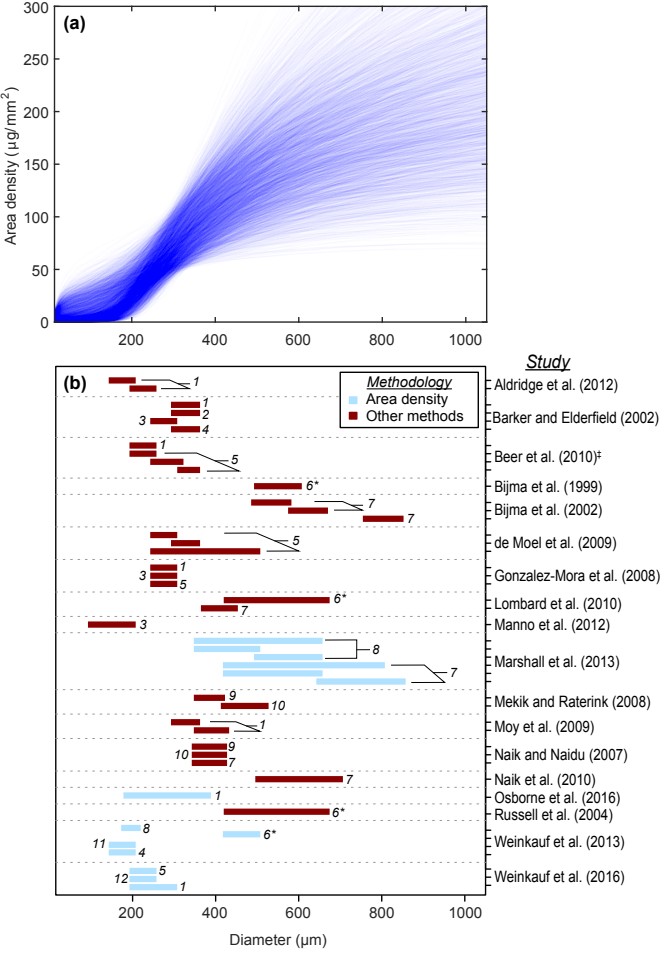

**Figure 6.** Modelled dependency of area density ($\rho_A$) on foraminiferal diameter (panel a), for comparison with the wide range of size fractions used in studies of size-normalised weight (panel b). Note we model area density in panel a, and so those studies that also use this exact metric are shaded separately in blue in panel (b). Species numbers are (1) *Globigerina bulloides*, (2) *Truncorotalia truncatulinoides*, (3) *Neogloboquadrina pachyderma*, (4) *Globoconella inflata*, (5) *Globigerinoides ruber* (white), (6) *Orbulina universa*, (7) *Trilobatus sacculifer*, (8) *Globigerinoides ruber* (pink), (9) *Neogloboquadrina dutertrei*, (10) *Pulleniatina obliquiloculata*, (11) *Globorotalia scitula*, (12) *Globigerinoides elongatus*. *Since *O. universa* is spherical, we stress that test size exerts a negligible control on $\rho_A$, in contrast to other species. ‡ We note that this study uses cross-sectional area in calculating a size-normalised weight, but their approach to normalisation to a set size is different to the $\rho_A$ method outlined in Marshall et al. (2013).





**Table 1.** Description of important terms used in this paper and the relevant literature.

| Term | Shorthand | Meaning | Further reading |
|---|---|---|---|
| Size-normalised weight | SNW | A general term for the mass of foraminiferal tests divided by some metric of test size. The term describes how 'heavily calcified' foraminiferal shells are, but importantly, it does not discern between changes in the degree of calcification during life vs. post-mortem thinning and dissolution. Many methods exist in the literature for normalising test mass to size, each with merits and pitfalls. | e.g. Aldridge et al. (2012); Beer et al. (2010a) |
| Calcification Intensity | CI | A more specific term under the umbrella of SNW that refers to how thickly the foraminifera calcified in life. For our culture experiments, this is defined more specifically in Eq. 2. | this study, Sect. 2.3 |
| Area Density | $\rho_A$ | A specific metric of size-normalised weight that normalises test mass to cross-sectional area. Since it has been increasingly used in recent studies, we discuss some implications for this particular metric. | e.g. Marshall et al. (2013); Weinkauf et al. (2016) |





**Table 2.** Data-constrained *G. ruber* measurements used to construct the model. †Variable names are those used to construct the Matlab code, available in the supporting material.

| Parameter | Variable name† | Value | Definition/notes | Source |
|---|---|---|---|---|
| Foraminifer aspect ratio | forAspRat | 1.16 | Test aspect ratio (major/minor axis) | This study |
| Foraminifer test growth rate | forAxIn | 1.19 | Fraction increase in test major axis per chamber addition | This study |
| Number of chambers | noCh | 16 | To better assess model behaviour 18 chambers were modelled, but only 16 were utilised in calculations and plots | Parker (1962) |
| Chamber aspect ratio | chAspRat | 1.66 | Chamber aspect ratio (major/minor axis) | This study |
| Chamber growth rate | axInOb | 1.15 | Fraction increase in chamber major axis per chamber addition | This study |
| Chamber overlap with previous | chCut | 45% | Proportion of spheroid chamber | This study |
| Initial chamber minor axis | inchAx2 | 5 $\mu$m | Proloculus semi axis based on *T. sacculifer* | Schmidt et al. (2013) |
| Porosity | forPor | 4.2% | Percentage chamber wall that is pore space | This study |
| Maximum primary wall thickness | maxInWall | | Saturation point of wall thickness with ontogeny: | |
| | | 10.5-19.5 $\mu$m | Model 2 | de Moel et al. (2009) |
| | | 18-22 $\mu$m | Models 1 and 3 | This study |
| Secondary calcite layer thickness | secChAd | 67% | Secondary calcite added per chamber addition, calculated as a proportion of the mass of all previous chambers | This study |





**Table 3.** Culture experiment details, raw calcification intensity (CI), and residual CI calculated following correction for the observed and modelled relationship between CI and axes product. Note for experiments beginning 'DE', carbonate ion concentrations are calculated with carbonate system constants adjusted for $Mg/Ca_{sw}$ using the MyAMI model (Hain et al., 2015). For MH2-Exp1, carbonate system calculations were adjusted for artificially enhanced seawater boron concentrations. DIC concentrations from experiments beginning 'DE' are from measured composite seawater solutions, and for experiments beginning 'MH' are derived from total alkalinity measured in composite seawater solutions.

| Experiment | Temperature (°C) | $Mg/Ca_{sw}$ (mol/mol) | pH (total) | ± | DIC ($\mu$M) | $[CO_3^{2-}]$ ($\mu$M) | CI ± SE | Initial product of axes (mm$^2$) | Mean chamber addition | Residual CI |
|---|---|---|---|---|---|---|---|---|---|---|
| DE3-2-26 | 26.3 | 2.17 | 8 | 0.1 | 2194 | 228 | 194±32 | 0.096 | 1.4 | -48 |
| DE3-3-26 | 26.3 | 3.25 | 8 | 0.1 | 2067 | 216 | 126±26 | 0.043 | 2.5 | -11 |
| DE3-4-26 | 26.3 | 4.15 | 8 | 0.1 | 2057 | 209 | 127±42 | 0.043 | 2.0 | 1 |
| DE3-6-26 | 26.3 | 6.25 | 8 | 0.1 | 2042 | 201 | 202±29 | 0.083 | 1.8 | 14 |
| DE4-3-22.5 | 22.8 | 3.40 | 8.2 | 0.1 | 2024 | 281 | 142±30 | 0.048 | 1.2 | 22 |
| DE4-3-25 | 25.3 | 3.40 | 8.2 | 0.1 | 1997 | 300 | 209±28 | 0.034 | 3.8 | 88 |
| DE4-3-27.5 | 27.8 | 3.40 | 8.2 | 0.1 | 2072 | 336 | 163±42 | 0.059 | 1.3 | 6 |
| MH1-HighpH | 26.0 | 5.18 | 8.182 | 0.007 | 1959 | 297 | 120±5 | 0.034 | 3.5 | -1 |
| MH1-MedpH | 26.0 | 5.18 | 7.893 | 0.013 | 1955 | 164 | 91±4 | 0.027 | 4.2 | -13 |
| MH1-LowpH | 26.0 | 5.18 | 7.564 | 0.008 | 1942 | 79 | 72±10 | 0.033 | 3.0 | -37 |
| MH2-Exp1 | 26.0 | 5.18 | 7.617 | 0.01 | 2064 | 94 | 97±47 | 0.050 | 2.2 | -47 |

## Appendix A: Model Description and Discussion

### A1 Model introduction

A number of excellent models already exist that describe chamber addition and three-dimensional coiling in foraminifera (e.g. Berger, 1969;
Signes et al., 1993). The models we present here, by contrast, are more directly focussed on addressing questions of foraminiferal size-normalised shell weight, which can be difficult to address with these more complex types of models. The annotated source Matlab code for the model accompanies this paper. With simplicity in mind, o ur models track the mass of calcium carbonate in a particular chamber, as the foraminifera grows. With each chamber addition, the amount of the total calcium carbonate in the shell is summed up, before the content of the subsequent chamber is calculated. These data can then be normalised to overall test size to calculate metrics of size-normalised weight
currently used in palaeoceanographic studies.

The mass of calcium carbonate in a particular chamber is determined from a number of prescribed morphological and wall-thickness parameters, which are derived from empirical observations. Morphological parameters used to determine a chamber's carbonate content include chamber size relative to preceding chambers, chamber aspect ratio (i.e., relative round or ovoid), and relative overlap with previous chambers (effectively hiding part of test). Wall thickness parameters include the thickness of the initial chamber wall (i.e. before subsequent
thickening- the 'primary wall'), wall porosity, and the thickness of secondary layers added to preceding chambers. Wall thickness was then scaled with ontogeny either as a function of size or of chamber number (see below), up to a parameterised maximum primary wall thickness that was constrained from observations of *G. ruber* (see Table 2, Supplementary Table 1, and Supplementary Figure 1).



Three distinct models were explored to assess the importance of the choice of growth model (illustrated schematically in Fig. A1):

- **Model 1**. Ontogenetic changes in calcification were modelled as a function of body size, with calcification changes controlled by the shape of the relationship between wall thickness and increasing chamber size.

- **Model 2**. As model 1, except that calcification changes through ontogeny were mainly driven by varying the maximum primary wall thickness attainable in later chambers.

- **Model 3**. As model 1, except that ontogenetic changes in wall thickness were varied as a function of chamber number rather than increasing chamber size.

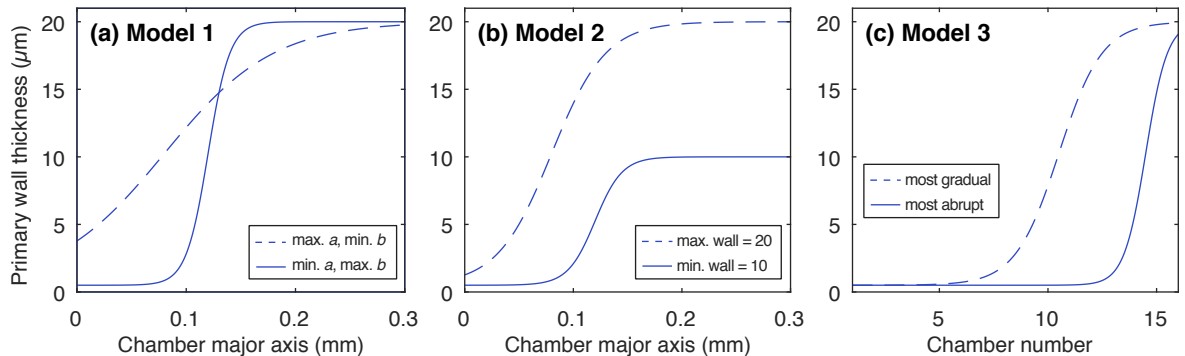

**Figure A1.** Scaling of wall thickness with chamber size in our models, showing the principle difference between the models described in Sect. 2.4.2. (a) Model 1 was designed to test the effect of varying the slope (parameterised by constants $a$ and $b$) of this relationship. (b) Model 2 was designed to test the effect of varying the maximum primary chamber wall thickness, while keeping $a$ and $b$ roughly constant. (c) Model 3 parameterises wall thickness as a function of chamber number instead of body size. The two lines in each model show the maximum extent by which this regression was allowed to randomly vary, and therefore delineate model extremes. See equation A2 for the definition of this relationship.

The differences between models appear subtle, but are important. In models 1 and 2, two post-gametogenic foraminifera of different sizes
would have a different calcite thickness on their final chamber – the larger individual would have thicker final chamber calcite. By contrast in model 3, two post-gametogenic foraminifera of different sizes, but the *same total chamber number*, would have the same calcite thickness on their final chamber. In a scenario where calcification is limited by availability of the necessary ions rather than energetic cost, model 1 foraminifera might accommodate unfavourable conditions by maintaining chamber dimensions at the expense of wall thickness, whereas model 3 foraminifera would maintain chamber wall thickness at the expense of chamber size and/or shape. Similarly, the difference between
models 1 and 2 is subtle but important. Both models explore the role that the relationship between foraminifer size and wall thickness exert on calcification intensity. However, model 1 achieves this mainly through varying the test size at which wall thickness increases begin to rapidly ramp up, whereas model 2 instead varies the maximum primary wall thicknesses attained in later chambers (although we stress that some flexibility was maintained in all parameters). Together, these models cover a range of possible scenarios for parameterising ontogenetic and environmental calcification change, and hence provide a solid framework for understanding the size-dependence of inferred CI and the
apparent conflict in previous studies.





## A2    Model construction

Each model was constructed on the basis of sequential addition of spheroid chambers of a set porosity. Chamber addition is initialised relative to a specified first chamber (i.e., proloculus) with an average diameter of 10 $\mu$m as based on empirical measurements (Table 2). Approximate

chamber volume was calculated by modelling chambers as spheroids:

$$\text{chamber volume} = 4/3\pi \times \text{semiaxis}_1 \times \text{semiaxis}_2^2 \tag{A1}$$

Where $\text{semiaxis}_1$ is half the Feret's diameter of the spheroid chamber, and is equal to $\text{semiaxis}_2$ multiplied by the prescribed chamber aspect ratio (Table 2). Chamber addition continued in steps until all chambers were added. *G. ruber* typically adds between 15 and 17 chambers (Parker, 1962), so a terminal chamber count of 16 was used in all models (Table 2).

Specific morphometric measurements from which the ontogenetic model was constructed are given in Supplementary Table 1 and Supplementary Figure 1. These are based on core-top *G. ruber* from the equatorial Atlantic; see main text Section 2.4.1 for full description. Because we observe no significant trend in any of these parameters as a function of size (Supplementary Figure 1), we use the mean of all measurements to derive the numbers stated in Table 2. Porosity measurements were on foraminifera spanning a smaller size range than those

from which the other measurements were taken (250-425 $\mu$m), so we cannot constrain possible ontogenetic changes in porosity based on this dataset.

The three important features of the model are: 1) ontogenetic growth and parameterisation with random variation; 2) independent variation of test size and chamber size; and 3) the inclusion and effect of non-linear ontogenetic changes. Firstly, with each step in ontogeny (i.e., each chamber addition) the amount of calcium carbonate added is determined as a function of body size in the preceding step (size-models, Models

1 and 2) or chamber number (chamber-model, Model 3) based on the empirical *G. ruber* measurements in Table 2 and Supplementary Table 1. Specifically, on average, each new chamber had a major axis length 15% greater than the preceding one, a chamber aspect ratio of 1.66 (major/minor chamber axis length), an overlap of 45% with previous chambers, and a wall porosity of 4.2%. With each chamber addition, the total mass of carbonate in the foraminifera was determined as a function of the size, shape, and porosity of the newly added chamber (parameters described above), the thickness of calcite wall of the new chamber (which varied ontogenetically according to the three models

listed above, described in detail below), and the addition of secondary calcite to the pre-existing test (on average parameterised as 67% of the cumulative carbonate content).

For a given model run, each parameter (i.e., model input) was varied randomly from the mean parameter listed in Table 2 by ±10%. This allowed us to explore parameter space and account for uncertainty in each biometric input. Whilst individual foraminifera may deviate by more than 10% from the population mean for any given parameter (see morphological measurements of *G. ruber* in Supplementary Table 1

and Supplementary Figure 1), we assume that the population mean (the parameter primarily of interest in calcification intensity studies) does not vary beyond this range.

Secondly, to calculate a size-normalised weight as in culture and field studies, the model requires a measure of total foraminiferal size (approximated as the sum of the maximum and perpendicular axes). We modelled this independently as a cumulative function, based on the observed increase in foraminifer diameter per chamber addition from our measurements. The test maximum axis (i.e. maximum Feret's

diameter) was increased by 19% on average per chamber addition, with an average test major/minor axis ratio of 1.16. In this way, the foraminiferal test area was not derived directly from the size of the modelled chambers. Instead, chamber size and foraminifer size were allowed to vary independently from each other. A benefit of this approach (besides computational efficiency) is that foraminiferal morphology varies in the third dimension (height), so by modelling chamber size and foraminifer size independently we effectively are capturing this

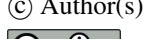



third dimensional variation. A limitation of this model structure is that it can produce morphologically impossible scenarios, for example foraminifer with a maximum test Feret's diameter less than that of the final chamber. We therefore used natural population measurements of *G. ruber* to filter model parameter combinations. Specifically, randomly generated model combinations were discarded if they a) produced

impossible morphological scenarios, like that described above or b) produced area-mass ratios far outside of that observed in natural *G. ruber* populations in the Gulf of Aqaba (Eilat). In the second case, models falling outside of root mean sq. error of 3.12 of the best-fit regression line of the Gulf of Aqaba populations were discarded. In practical terms, this allows for up to a factor of two change in the size-mass relationship given in Eq. 1. This range in successful models constitute ∼8% of the total runs for each model type.

A third key feature of this model is the inclusion of non-linear ontogenetic changes in calcification. Many aspects of foraminiferal morphology, including chamber dimensions and coiling, primary wall thickness, and porosity are known to change non-linearly over ontogeny. Brummer et al. (1986, 1987) noted distinct phases in foraminiferal morphology correlated with chamber count, a finding supported by new results from Schmidt et al. (2013). For *G. ruber*, we lacked the in-depth ontogenetic measurements available for other modern species (Schmidt et al., 2013), and so we parameterise all non-linear growth functions as non-linear primary wall thickness functions (described below; note

for clarity, we define primary wall thickness as the thickness of a given chamber wall after precipitation but prior to any secondary calcification during later chamber additions). We recognise that similar transitions likely occur in other aspects of morphology in step with changes in wall thickness, but since wall thickness is likely volumetrically to be by far the most important (rather than e.g. differential porosity in early chambers), it seems reasonable to essentially treat primary wall thickness as a proxy for all such conflated transitions. Our empirical measurements of *G. ruber* wall thickness (Table 2, Supplementary Table 1) reveal that there is no relationship between wall thickness and

test diameter above a diameter of ∼ 250 $\mu$m, with wall thickness remaining roughly constant at ∼20 $\mu$m. We therefore define three logistic relationships to describe the co-variation of primary chamber wall thickness with foraminifer size (shown schematically above in Fig. A1)), such that:

$$\text{primary chamber thickness} = \frac{\text{maximum primary thickness}}{1 + \exp(a(x - b))} \tag{A2}$$

where $a$ and $b$ are constants, and $x$ is a measure of test size- either the previous chamber's major axis (models 1 and 2) or chamber number

(model 3). The coefficients that define how wall thickness increases over ontogeny describe the shape of a non-linear relationship. The first coefficient ($a$) controls how tightly curved the regression is; models with more negative values of $a$ have thinner chambers during earlier stages of growth, but then more quickly transition to growing chambers with maximum primary wall thickness. The second coefficient ($b$) defines the size at which the foraminifera begin to build thicker chambers, i.e. models with lower values of b reach maximum primary wall thickness is reached at a lower chamber diameter (Supplementary Figure 2).

Maximum primary wall thickness in models 1 and 3 is fixed at 20 $\mu$m ($\pm$10%) on the basis of our core-top measurements of *G. ruber*. In model 2, to test the influence of this parameter, a and b were constrained to within 10%, but maximum test wall was allowed to vary more widely, between 10-20 $\mu$m. In models 1 and 3, the shape of the logistic regression (as described by $a$ and $b$ in Eq. A2 above, and illustrated in Supplementary Figure 2) was allowed to vary considerably as, to our knowledge, it is unconstrained by observation. Limits were set only

by the post-hoc screening of models that fell outside of the possible range of natural populations (for example, those in which proloculus wall thickness is greater than the chamber diameter).



Because this simple model can be run quickly, we could allow model parameters to vary randomly (within tolerances) and independently of each other, so as to interrogate the parameters driving CI. As stated in the main text, any individual parameter can drive calcification intensity if varied in isolation and unconstrained by the requirement of an ontogenetic model to approximate size-mass relationships seen in the open-ocean samples. However, when we examine only the subset ($\sim 8\%$) of model permutations that produce realistic size-weight relationships, no significant relationship between CI and any one parameter listed in Table 2 is observed. This is demonstrated in Supplementary Figure 3 by plotting the relationship between modelled CI against all input parameters (at a body size of $\sim 0.1$ mm$^2$, for the case of two chamber additions). None of these parameters alone drives CI, which as we discuss in the main text, has the implication that a change in one may be offset by a change in another (that shifts CI in the opposite direction) in order to produce foraminifera that conform – within tolerances – to the observed size-weight relationship for this species. In contrast, the parameterisation ($a$ and $b$) of the relationship between wall thickness and body size or chamber number unavoidably drive CI irrespective of simultaneous changes in the other parameters. This is the case irrespective of whether primary wall thickness is defined as a function of chamber diameter (model 1) or chamber number (model 3) (Fig. A2, see also Sec. 2.4.2). However, changing the maximum wall thickness that is reached at maturity while keeping a and b fixed to within $\pm 10\%$ (model 2) has no significant effect on CI. The dependence of CI on coefficients a and b in isolation is also far less strong than when both vary in tandem (Figure A2, right hand panels). This suggests it is the overall shape of the regression between wall thickness and ontogeny, rather than just one of its constituent coefficients, which is important in controlling CI – thereby leading us to our hypothesis of size-dependent calcification responses.





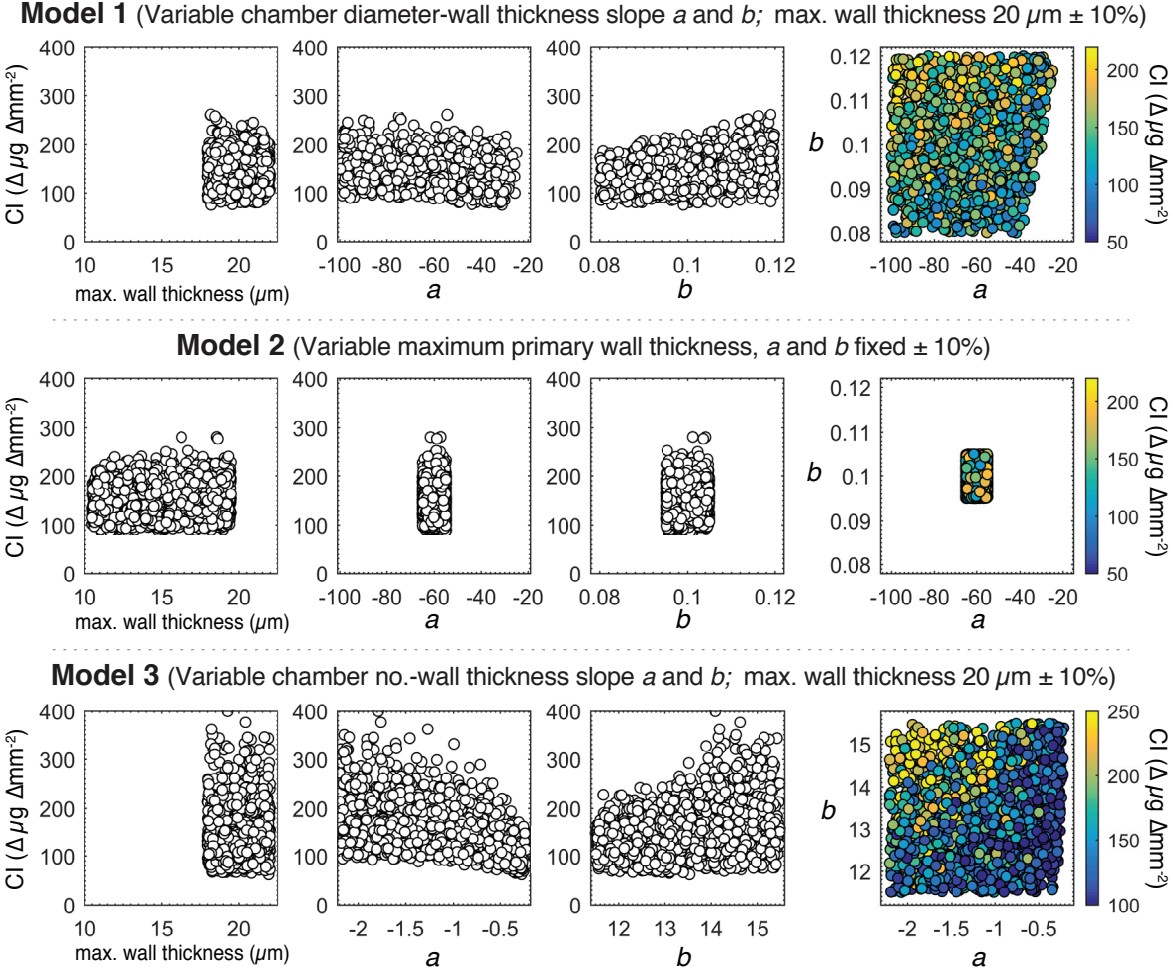

**Figure A2.** Modelled dependence of CI on morphological parameters within the three model groups for foraminifera of $\sim$0.1 mm$^2$. No parameter exerts a systematic control on CI in foraminifera of this size with the exception of the coefficients $a$ and $b$ that parameterise the relationship between body size and primary wall thickness (see Eq. A2). In Model 1 (top row), these coefficients are varied, but the maximum primary wall thickness (i.e. before secondar thickening) is kept at $20 \pm 2$ $\mu$m. In model 2, conversely, $a$ and $b$ are constrained, with maximum chamber thickness allowed to vary. In model 3, wall thickness is scaled with at chamber addition step, rather than chamber size. Regardless of base model, varying $a$ and $b$ will drive CI changes (bottom and top row, central pairs). Varying maximum wall thickness does not drive such a relationship (middle row left). When changed together, the $a$ and $b$ explain $\sim$50% of the variance in CI, as illustrated by the coloured plots (models 1 and 3, top- and bottom-right panels).

## A3    Model caveats

An inherent limitation in our models is that parameter sets are not utilised if they do not produce realistic mass-size curves matching the natural mass-axes product relationship of *G. ruber* in the Red Sea. However, a substantial relaxation of this tolerance does not significantly





change our results. Moreover, relaxing this tolerance too far would result in mass-size relationships that can no longer be reasonably assumed to represent the species *G. ruber*. Although we note that the tolerances permitted for matching the natural populations are quite large (they allow for change in the size-mass relationship by a factor of two), this natural population itself was sampled from within a narrow range in

ambient pH. As a result, we cannot unequivocally rule out a pH-induced change in one of the input parameters in isolation to change CI in response to acidification. That said, to our knowledge there is little empirical support for factors such as chamber aspect ratio or porosity to respond drastically to acidification. In addition, in the open ocean, trade-offs in allocation of calcification resources likely operate that make it difficult for one morphological parameter alone to drive CI. For example, a decrease in the thickness of secondary calcite layering might be compensated for by building smaller chambers so as to ensure structural integrity is maintained. Similarly, the need for cellular

defence would preclude an increase in porosity of the magnitude that would be required to effect large changes in CI. Thus we suggest that allowing all parameters to vary randomly and then screening models may be more realistic in reproducing morphological variability in natural populations.

Another potential limitation of this model may stem from our parameterisation of shell thickening as occurring as a series of discrete

additions concurrent with each chamber formation (*sensu* Erez, 2003). Emerging findings from species such as *Neogloboquadrina dutertrei* (e.g. Fehrenbacher et al., 2016) suggest that at least some foraminifera may first add chambers until they reach a final test size, and then subsequently thicken all chambers continually over some days prior to gametogenesis. Note that this is distinct from concepts such gametogenic calcite addition or encrustation, and refers specifically to ontogenetic thickening. It has been argued that the same continual thickening processes are observed in *Orbulina universa* (Spero et al., 2015). However, to date, there has been no investigation confirming the exis-

tence of such a thickening mechanism in *G. ruber*. Thus, while future work could yet reveal some secondary thickening process at work, at present we lack the observational constraints required to incorporate such a mechanism in our models. In a practical sense, however, we suspect the conceptualisation of shell thickening chosen is not likely to greatly impact our conclusions, for two reasons. Firstly, even with observed end-stage thickening in *N. dutertrei*, older chambers are often more heavily thickened than later chambers (Fehrenbacher et al., 2016), which could in a post-hoc sense result in similarly thickened older chambers as in the models we use here, even if the ontogenetic

pathway to achieving this differs. Secondly, and perhaps more crucially, our model ontogenies are screened against size-weight relationships in towed foraminifera, and so by necessity our modelled ontogenetic trends must approximate true physiology. Nonetheless, should secondary thickening be observed in *G. ruber* in future, these sorts of modelling exercises should be revisited.