# Peer review of "Size-dependent response of foraminiferal calcification to seawater carbonate chemistry"

_Biogeosciences, 2016_

## Referee Comment (RC1) · R. Schiebel (Referee) · 29 Nov 2016

The paper of Henehan and coauthors on 'Size-dependent response of foraminiferal calcification to seawater carbonate chemistry' presents a nice new approach to better understand the formation of planktic foraminifer shell calcite, its use as a proxy in paleoclimate research, and possible feedbacks to rising atmospheric and surface ocean carbon dioxide concentration. In general, the manuscript is written in a clear way, and statements are unequivocal. In the following, I comment on three points, meant to enhance intelligibility of the paper.

First point: On page 8, lines 9-11, Henehan et al. 'Note that logarithmic regression models were used because modelled CI through ontogeny approximates to a logarithmic relationship across the size range of our cultures (see Fig. 1b).' It should

be made clear whether 'cultures' only include the size-to-calcite mass relationships of entire individuals of assemblages, or if data on the ontogenetic development (from cross-sections, or CT) of single specimens where also included here.

Second pointWhen comparing their data to the published data of others, Henehan et al. seem to have struggled with the classification of G. ruber morphotypes, and a general confusion concerning taxonomy of G. ruber as presented in the literature. To my knowledge, Wang (2000) first described different water depth habitats of G. ruber s.s. and s.l. from the South China Sea (SCS). Wang (2000) knew all about the difference between the different morphotypes (elongatus and pyramidalis), but finally only used differentiated between the types with spherical final chambers (s.s.) and compressed final chambers. From Wang (2000): 'Initially, the Globigerinoides ruber s.l. group was differentiated into tests with low and high trochospires. However, as these two subgroups did not show significant differences in their isotopic signal, they were lumped again into one group.' The morphotype with the compressed final chamber is referred to as platys by some colleagues (see Numberger et al. 2009), and may just represent specimens with a kummerform final chamber. The concept of Wang (2000) was then largely adopted by Steinke et al. (2005) also working on the SCS and Indo-Pacific waters. Beer, Schiebel, Wilson (2009) did certainly distinguish between the different morphotypes, and did only use G. ruber (white), i.e. G. ruber s.s., in their analyses. G. ruber, G. elongatus, and G. pyramidalis were considered different species. However, tests with normal formed and kummerform final chamber were no distinguished, because we the size-to-weight ratio of these tests was not significantly different in the samples from the Arabian Sea. Please note: The ecological significance of different morphotypes, i.e. G. ruber s.l. in warmer waters, and G. elongatus and G. pyramidalis in colder waters as found by Steinke et al. (2005) may differ at the regional scale. Water temperature may just be one among many (more relevant?) parameters, which determine the ecological niche of a species. To conclude, the statement on p. 10, lines 23-25, is wrong, and should be corrected: 'Beer et al. did not differentiate between species, but it is likely they would have sampled an increasing proportion of

higher-SNW sensu lato species (i.e. G. elongatus and G. pyramidalis) in lower-pH up-welling waters, given these species preference for colder waters (Steinke et al., 2005). I would have happily discussed this point with the authors before submission of your manuscript, and I might have even provided you with the original samples.

Third point: In Figure 5, Henehan et al. show present a schematic view of the factors affecting shell thickness, by comparing large modern planktic foraminifers and small Paleocene-Eocene benthic foraminifers. To my consideration, this is comparing apples and oranges, and is hence insignificant. Calcification in benthic foraminifers is possibly related to the nature and chemistry of the bulk sediment, and follow an entirely different systematics than in planktic foraminifers. Natural and cultured specimens "pH reaction" may just reflect the general health of individuals, which might be related to alimentation. In addition, production and preservation may both affect wall thickness: Pores in the images (Fig. 5) of G. ruber are funnel shaped, which may indicate dissolution. I would suggest to change Fig. 5 and text on page 9 and 10. By the way: The expression 'Larger Foraminifera' signifies an informal group of large benthic foraminifers, and should not be used for planktic foraminifers and other benthic foraminifers, to avoid confusion.

---

## Author Comment (AC1) · 10 Feb 2017

*Reply to Referee comment 1 from R. Schiebel on* "Size-dependent response of foraminiferal calcification to seawater carbonate chemistry" *by* Henehan, Evans et al.

We thank the reviewer for taking the time to review our manuscript, and for his positive and constructive comments. Below we detail the specific responses to the three points made.

**First comment:** *"On page 8, lines 9-11, Henehan et al. note that 'logarithmic regression models were used because modelled CI through ontogeny approximates to a logarithmic relationship across the size range of our cultures (see Fig. 1b).' It should be made clear whether 'cultures' only include the size-to-calcite mass relationships of entire individuals of assemblages, or if data on the ontogenetic development (from cross-sections, or CT) of single specimens where also included here."*

**Response:** We fear that our writing was perhaps not clear enough here. What we mean to say is that we fit a logarithmic regression to the culture results (i.e. solely the results of our culture experiments, and nothing else) because the models that we have constructed indicate that the increase in calcification intensity with size over the size range of our cultures is approximately logarithmic. To illustrate this, in the attached figure 1 below we shade the range of sizes seen in culture (plots from Main text Fig. 4), to compare with the models (from Main text Fig. 2).

These models are built from our own measurements and published measurements from techniques such as synchrotron radiation X-ray tomographic microscopy (see Table 2), and are then screened by comparison with open-ocean Red Sea populations (see Section 2.4.2 and Appendix A). To reiterate, the objective here is only to describe the underlying size control on CI, so as to then determine the residual variability caused by pH change. We therefore use the shape of a relationship between CI and size ( logarithmic over this size range) derived from our models (itself built on cross-sections, etc.), to inform as to the most appropriate fit. We will clarify the revised text, and also include clarification on the morphospecies used in cultures and in the construction of the model.

**Second point (broken down for clarity):** *"When comparing their data to the published data of others, Henehan et al. seem to have struggled with the classification of* G. ruber *morphotypes, and a general confusion concerning taxonomy of* G. ruber

*as presented in the literature. To my knowledge, Wang (2000) first described different water depth habitats of* G. ruber *s.s. and s.l. from the South China Sea (SCS). Wang (2000) knew all about the difference between the different morphotypes (*elongatus *and* pyramidalis*), but finally only used differentiated between the types with spherical final chambers (s.s.) and compressed final chambers. From Wang (2000): 'Initially the* Globigerinoides ruber *s.l. group was differentiated into tests with low and high trochospires. However, as these two sub-groups did not show significant differences in their isotopic signal, they were lumped again into one group.' The morphotype with the compressed final chamber is referred to as* platys *by some colleagues (see Numberger et al. 2009), and may just represent specimens with a kummerform final chamber. The concept of Wang (2000) was then largely adopted by Steinke et al. (2005) also working on the SCS and Indo-Pacific waters. Beer, Schiebel, Wilson (2009) did certainly distinguish between the different morphotypes, and did only use* G. ruber *(white), i.e.* G. ruber *s.s., in their analyses.* G. ruber, G. elongatus, *and* G. pyramidalis *were considered different species. However, tests with normal formed and kummerform final chamber were no distinguished, because we the size-to-weight ratio of these tests was not significantly different in the samples from the Arabian Sea.*

**Response:** We thank the reviewer for the comments regarding our use of species concepts with regards *G. ruber* in the published literature, and have reviewed the manuscript for locations where we could be clearer and more exact. We will clarify our taxonomy in multiple locations of the main text, including the caption to figure 4, to address this concern.

*Please note: The ecological significance of different morphotypes, i.e.* G. ruber *s.l. in warmer waters, and* G. elongatus *and* G. pyramidalis *in colder waters as found by Steinke et al. (2005) may differ at the regional scale. Water temperature may just be one among many (more relevant?) parameters, which determine the ecological niche*

*of a species.*

**Response:** While we agree that there may be other controls on the relative abundance of *G. ruber* s.l., certainly the geographic range of these morphotypes may suggest temperature is a very important control. For instance, in core tops we have worked with characterised by mean annual surface water temperatures of 15-20 °C in the North Atlantic and Southwest Pacific, it is common to find *G. ruber* s.l., but none of the more tropical sensu stricto species. A paper published this week also indicates that sensu lato *ruber* is associated with colder waters than co-habiting sensu stricto in the East China Sea (A. Carter et al., Marine Micropalaeontology, doi:10.1016/j.marmicro.2017.01.001). In that sense it seems a reasonable null hypothesis that *G. ruber* s.l. either tolerates, or even prefers, colder temperatures, even if there may of course be other factors at work.

*"To conclude, the statement on p. 10, lines 23-25, is wrong, and should be corrected: 'Beer et al. did not differentiate between species, but it is likely they would have sampled an increasing proportion of higher-SNW sensu lato species (i.e.* G. elongatus *and* G. pyramidalis*) in lower-pH upwelling waters, given these species preference for colder waters (Steinke et al., 2005).' I would have happily discussed this point with the authors before submission of your manuscript, and I might have even provided you with the original samples."*

**Response:** We agree that our communication could have been better, and apologise for this oversight. We will remove this passage from the revised discussion altogether, since we have no alternative explanation to offer for the disagreement between our data and those of Beer et al. (2010).

**Third point (broken down for clarity):** *"In Figure 5, Henehan et al. show present a schematic view of the factors affecting shell thickness, by comparing large modern planktic foraminifers and small Paleocene-Eocene benthic foraminifers. To my consideration, this is comparing apples and oranges, and is hence insignificant. Calcification in benthic foraminifers is possibly related to the nature and chemistry of the bulk sediment, and follow an entirely different systematics than in planktic foraminifers."*

**Response:** While we appreciate that benthic and planktic foraminifera have clear eco-physiological differences, we would argue that there is no reason to think that their mechanisms of calcification are fundamentally different. Much of our understanding about biomineralisation processes in foraminifera comes from laboratory studies in benthic foraminifera (e.g. ter Kuile and Erez 1988, 1989; Elderfield et al., 1996; Bentov and Erez, 2005; Bentov et al., 2009; de Nooijer et al., 2009; de Nooijer et al., 2014), and conclusions from these benthic foraminifera are extended often, if not always, to the rest of the non-porcelaneous calcareous foraminifera. In addition, observations from benthic foraminifera we discuss here are seen in both epifaunal (*N. truempyi*) and shallow-infaunal (*O. umbonatus*) taxa, suggesting porewater chemistry is not the driving factor here (Foster et al., PNAS 2013). Furthermore, it was recently observed that the proton flux associated with calcification in the shallow-dwelling benthic foraminifera *Ammonia* does not change as a function of seawater pH (Toyofuku et al., 2017 Nat. Comms.). This observation, from a very different benthic species, is in agreement with our model in that it also indicates a limited response of calcification to reduced pH in small species that probably lack an internal calcium or carbon pool.

We therefore do not think it unreasonable to posit a hypothetical model regarding commonality in calcification behaviour as we do, so that these ideas can be tested and falsified. However, we recognize the uncertainties associated with this comparison, and will add this caveat clearly to the revised text, while emphasising that this is as yet an untested hypothesis.

*"Natural and cultured specimens 'pH reaction' may just reflect the general health of individuals, which might be related to alimentation."*

**Response:** We stress that our cultured foraminifera were all offered food at the same rate between pH experiments, and so alimentation should not have a role in producing our observed pH effect in culture. Furthermore, we would suggest the findings of Aldridge et al. (2012) in *Globigerina bulloides* indicate food supply has no greater influence on shell weight than co-varying carbonate system parameters.

*"In addition, production and preservation may both affect wall thickness: Pores in the images (Fig. 5) of* G. ruber *are funnel shaped, which may indicate dissolution. I would suggest to change Fig. 5 and text on page 9 and 10.*

**Response:** We agree that preservation can be an issue when interpreting sediment records- a point we make firmly in the manuscript based on our analysis of core-tops. However, in the case of this figure, de Moel et al (2009) considered the possibility of dissolution, and regard it as unlikely: *"The shells in the sediment ... generally look well preserved, some with remnants of spines still present. Fragmentation and dissolution are known to change faunal assemblages (Berger, 1970; Anderson and Archer, 2002; Le and Thunell, 1996), and susceptibility for it is related to the thickness of the shell walls (Barker et al., 2007). However, Conan et al. (2002) showed that exactly at this site the abundance of dissolution-sensitive species in the surface sediment is high and there is a close similarity between foraminifera assemblages and skeletal group compositions in the surface sediment and in an on-site sediment trap. This implies a good preservation without selective removal of susceptible carbonate components (i.e. thin walled shells) in the sediment."* From our experience in cultures, pores in

foraminifera grown in low pH conditions can appear slightly larger (see e.g. images in Henehan et al. 2013, Fig. 4), and so there is the potential that widened pores such as those in de Moel et al.'s figure could conceivably be due to some influence of pH on pore size. However, our culture observations are by no means quantitative. The question of controls on porosity is an active research question amongst our group, however, and so we hope to address this question in due course.

For these reasons, we respectfully prefer to retain the figures and text as presented, but with some rewording to the text and the explicit recognition that this remains a hypothetical model that should be tested by the community going forward.

*"By the way: The expression 'Larger Foraminifera' signifies an informal group of large benthic foraminifers, and should not be used for planktic foraminifers and other benthic foraminifers, to avoid confusion."*

**Response:** We will change the wording in the revised manuscript to 'bigger' rather than 'larger' so as to avoid any possible confusion.

[Figure]

**Fig. 1.** Clarification of the 'size range seen in culture' that we refer to in the text.

---

## Referee Comment (RC2) · Anonymous Referee #2 · 13 Feb 2017

In this manuscript, the authors use a combination of laboratory culture experiments, plankton tows, fossil shells and modeling to examine drivers of size normalized weight in foraminifera. They focus efforts on the species G. ruber, and identify the importance of shell size and chamber number as predictors of size normalized weight, while also reaffirming the roll of carbonate chemistry, both during growth and in the depositional environment. This paper represents an important contribution to understanding the mechanisms of and interpreting differences in foraminiferal weight in the fossil record. Overall the paper is well structured and well written. I have highlighted below a few areas where the authors make some broad assumptions in their reasoning, which if addressed directly, could further strengthen this manuscript.

1) Henehan et al. have, in their discussion, put forward some interesting ideas about the mechanisms underlying observed trends in calcification intensity in different sized

[Figure]

G. ruber. However, the extrapolation of this to all foraminifera (small/large, plank-tonic/benthic, juvenile/adult) is in my opinion overreach. This line of reasoning seems to equate adult foraminifera from small species with earlier ontological stages in larger species. However, it is unlikely that size alone is a meaningful determinant of physiology and calcification mechanisms across such a diverse group of foraminifera and ontological stages. I would recommend that the authors either remove these sections on pages 9-10 (and Fig. 5) or rework this discussion to better support and address these assumptions.

2) The novel approach presented in the methods for quantifying calcification intensity in cultured foraminifera could be widely used, but raises some questions. This metric relies on the assumption that foraminifera of a single species from a certain locale will have a consistent size/mass relationship, such that an initial mass can be predicted from size. However, the authors show that environmental conditions (carbonate chemistry) can significantly alter the size/mass relationship. This would seem to contradict the underlying assumption of consistent initial size/mass. This apparently contradiction could be made explicit and addressed.

For example: Was anything done to constrain the environmental conditions of the foraminifera used to establish an initial size/mass relationship? How do the conditions at collection of these samples compare to those at the collection of cultured foraminifera? The R2 of the initial relationship is also not very high (0.61), suggesting quite a lot of variability in individual foraminifera size/mass – could Henehan et al. give an indication of the uncertainty this would introduce into the calculation of calcification intensity in a cultured foraminifera?

Minor: The authors show that size-dependent calcification intensity is responsive to carbonate chemistry. Given this, they may wish to add an acknowledgment or brief discussion of the existing literature on how various environmental factors, like temperature, can impact shell size and growth rate (e.g. Schmidt et al., 2004 or Lombard et al., 2010).

Page 13, Line 14: edit "change changes"

---

## Editor Comment (EC1) · L.J. de Nooijer (Editor) · 23 Feb 2017

Dear Dr Henehan and co-authors,

I would like to invite you to reply to the points raised by the second reviewer and upload an improved version of your manuscript. I am looking forward see an updated version!

Sincerely,

Lennart de Nooijer

---

## Author Comment (AC2) · 6 Apr 2017

*Reply to Anonymous Referee comment 2 on* "Size-dependent response of foraminiferal calcification to seawater carbonate chemistry" *by* Henehan, Evans et al.

We thank the reviewer for some useful and constructive comments. Our responses to each are outlined below.

**Comment:** *"In this manuscript, the authors use a combination of laboratory culture experiments, plankton tows, fossil shells and modelling to examine drivers of size normalized weight in foraminifera. They focus efforts on the species* G. ruber *, and identify*

*the importance of shell size and chamber number as predictors of size normalized weight, while also reaffirming the roll of carbonate chemistry, both during growth and in the depositional environment. This paper represents an important contribution to understanding the mechanisms of and interpreting differences in foraminiferal weight in the fossil record. Overall the paper is well structured and well written. I have highlighted below a few areas where the authors make some broad assumptions in their reasoning, which if addressed directly, could further strengthen this manuscript.*
*"Henehan et al. have, in their discussion, put forward some interesting ideas about the mechanisms underlying observed trends in calcification intensity in different sized* G. ruber. *However, the extrapolation of this to all foraminifera (small/large, planktonic/benthic, juvenile/adult) is in my opinion overreach. This line of reasoning seems to equate adult foraminifera from small species with earlier ontological stages in larger species. However, it is unlikely that size alone is a meaningful determinant of physiology and calcification mechanisms across such a diverse group of foraminifera and ontological stages. I would recommend that the authors either remove these sections on pages 9-10 (and Fig. 5) or rework this discussion to better support and address these assumptions."*

**Response:** In light of the concerns of the reviewer, and the similarity to those concerns also expressed by reviewer 1, we have reworked this section of the manuscript. Specifically, we have:

- Removed the benthic foraminifera from Fig. 5 so as to reduce the emphasis on commonality of benthic and planktonic foraminiferal biomineralisation behaviour.

- Restructured the discussion in this section to more clearly separate hypothesis from subsequent treatment of empirical support.

- Explicitly stated the distinction between juvenile individuals and adult individuals of small species, and highlighted that at this time it is unclear to what extent these

two groups may be considered analogous in terms of calcification behaviour.

**Comment:** *"The novel approach presented in the methods for quantifying calcification intensity in cultured foraminifera could be widely used, but raises some questions. This metric relies on the assumption that foraminifera of a single species from a certain locale will have a consistent size/mass relationship, such that an initial mass can be predicted from size. However, the authors show that environmental conditions (carbonate chemistry) can significantly alter the size/mass relationship. This would seem to contradict the underlying assumption of consistent initial size/mass. This apparently contradiction could be made explicit and addressed."*

**Response:** The reviewer is indeed correct that environmental conditions can likely alter the relationship between size and mass. It is true that at other locations the size-mass relationship we observe may not be valid, and so we add the explicit recommendation that the relationship between size and mass be verified and/or recalibrated at new culture locations before attempting to use this metric (Section 2.3, Page 5, Lines 5-7).

**Comment:** *"For example: Was anything done to constrain the environmental conditions of the foraminifera used to establish an initial size/mass relationship? How do the conditions at collection of these samples compare to those at the collection of cultured foraminifera?"*

**Response:** Our non-cultured samples used to devise a size/mass relationship were taken from numerous tows from the Gulf of Eilat over the course of several years, and with each tow open ocean Eilat seawater was sampled for pH measurement at or close to the site of towing. Despite temporal variability, the tows fall within a narrow range of ocean pH 8.10 $\pm$ 0.05 (2se). This is close to the midpoint of the pH range of

our culture experiments. What's more, these individuals were pooled from the same tows that yielded the individuals that went into culture, and so there should not be any significant difference between the conditions at collections for culture vs. those in the size-mass calibration. Therefore, the findings from our culture experiments are robust. We now make these points in the manuscript Section 2.3, Page 5, Lines 2-7).

**Comment:** *"The $R^2$ of the initial relationship is also not very high (0.61), suggesting quite a lot of variability in individual foraminifera size/mass - could Henehan et al. give an indication of the uncertainty this would introduce into the calculation of calcification intensity in a cultured foraminifera?"*

**Response:** The reviewer is correct in the assertion that there is some scatter around our open-ocean size-mass relationship. However we stress that much of this scatter is likely to have arisen from measurement error, rather than true physiological variability. In particular, instrumental uncertainty on microbalance measurements is large in proportion to absolute shell mass. However, the absolute measurement uncertainty is independent of either variable, and the sample number is so large, the regression line itself is likely robust. Importantly also, by definition, our regression relationship is structured so that variability in the tow data-points is normally distributed around our line. Therefore there should be no systematic bias introduced into our culture CI data.

We do recognise that the scatter in the prediction intervals, if propagated through each individual tests' CI measurements, would produce a sizeable range in individual tests' CI- particularly for those foraminifera that did not add much mass in culture. However, for a number of reasons we suggest this shouldn't detract from the main findings of the paper. Firstly, the regression line is applied equally to all individuals and all experimental pH treatments, and so given that the error in the regression should be non-systematic, the foraminifera used in the size-mass relationship were towed at the

same time as culture specimens, and the water they were towed from was towards the midpoint of our culture pH treatments, relative changes between experiments should be robust. Secondly, each pH experiment consists of a combination of several individual CI datapoints within an experimental treatment, and so provided the sample size is large enough, the error on each individual test's CI calculation introduced from the size-mass calibration is averaged out on the treatment level. Since we recognised that the sample size within each treatment has a large effect on the uncertainty of each treatment (given the regression error and inter-individual variability) we assign error bars on our culture experiments based on sample size. We calculated this uncertainty by repeatedly subsampling smaller sets of individual foraminifera from one of our larger experiments with > 100 individuals, and noting the deviation of the mean of each subset from the true mean value (see attached Figure). Therefore the bounds of uncertainty given in the paper do incorporate the uncertainty stemming from interindividual variability and scatter in the size mass calibration.

The models that we built to simulate CI change through ontogeny are also grounded with this same size-mass relationship. Because there is some considerable scatter around this relationship, as the reviewer states, we allowed our models to vary within a root mean sq. error (RMSE) of 3.12 around this observed relationship. This permitted variability in modelled size weight relationship of up to approximately twice that seen in our sampled natural population. Therefore the conclusions drawn from our model relationships stand even when considering the large residual scatter in the tow measurements.

**Comment:** *"Minor: The authors show that size-dependent calcification intensity is responsive to carbonate chemistry. Given this, they may wish to add an acknowledgment or brief discussion of the existing literature on how various environmental factors, like temperature, can impact shell size and growth rate (e.g. Schmidt et al., 2004 or Lombard et al., 2010)."*

**Response:** We have now acknowledged this on Page 13, lines 6-8.

**Comment:** *"Line 14: edit "change changes""*

**Response:** Change changes changed.

**Figure Caption:** The relationship used to calculate our bounds of uncertainty. With repeated subsamples from a population of cultured individuals, the deviation of the mean of that subsample from the true mean can be calculated. This allow us to consider the effect sampling small numbers from a population with a large degree of inter-individual variability.
* * *
[Figure]

**Fig. 1.** Calculation of uncertainty from sample size.

---

## Author Response (AR1)

**Size-dependent response of foraminiferal calcification to seawater carbonate chemistry**

Michael J. Henehan[1], David Evans[1,2], Madison Shankle[1], Janet Burke[1], Gavin L. Foster[3], Eleni Anagnostou[3], Thomas B. Chalk[3], Joseph A. Stewart[3,4], Claudia H. S. Alt[3,5], Joseph Durrant[3], and Pincelli M. Hull[1]

[1]Department of Geology and Geophysics, Yale University, 210 Whitney Avenue, New Haven CT-06511, USA
[2]Department of Earth Sciences, University of St. Andrews, Irvine Building, North Street, St. Andrews, Fife, KY16 9AL, UK
[3]Ocean and Earth Science, University of Southampton, National Oceanography Centre Southampton, Southampton, SO14 3ZH, UK
[4]National Institute of Standards and Technology, Hollings Marine Laboratory, 331 Ft. Johnson Road Charleston, SC-29412, USA
[5]Department of Biology, College of Charleston, Charleston SC-29424, USA

*Correspondence to:* Michael Henehan & David Evans (michael.henehan@yale.edu; de32@st-andrews.ac.uk)

NOTE: Changes in response to Reviewer 1 are marked in red, while changes in response to Reviewer 2 are shown in blue. Line numbers for changes in response to Reviewer 2 are given in our Reply to their comment. For Reviewer 1, see (Line numbers in the resubmitted manuscript, not this annotated version):

Page 4, Line 20

Page 7, Line 7, Line 14

Page 8, Lines 10-12, Line 30, 31

Page 9, Line 31 - Page 10, Line 25

Page 10, Lines 28-29, 30-31

Page 13, Line 21, Line 22

[revised manuscript text omitted]

---

## Author Response (AR2)

***Reply to Editor's comments from L. de Nooijer on*** "**Size-dependent response of foraminiferal calcification to seawater carbonate chemistry**" ***by*** **Henehan, Evans et al.**

**Comment:** *"page 2, lines 24-27: I don't understand what 'conditions during life' means here. Don't these conditions include inorganic carbon chemistry at the sea surface (lines 24-25)? Or do you mean that 'conditions during life' within a foraminifer's lifetime are assumed to remain constant?"*

**Response:** What we mean here is that using SNW metrics as solely a dissolution index must assume relatively little influence of surface water conditions on size-normalised shell weight during the foraminifera's lifetime, such that the post-mortem signal reflects only dissolution. Conversely, using SNW as a proxy for surface water conditions requires post-depositional alteration to be minimal. We have rewritten this passage in case our writing was not clear.

**Comment:** *"page 3, line 11: 'Be' should be 'Bé'. Please also check other references to Bé in the rest of the manuscript."*

**Response:** Thank you for spotting this oversight. This has been corrected.

**Comment:** *"page 3, first paragraph: perhaps it is useful to include somewhere in the introduction the notion that SNW is often thought of/ assumed to reflect the 'average thickness of the chamber walls'. I would say that this may indeed be largely true, but in-/ decreased porosity (Bé et al., 1976. Progress in Micropaleontology 1-9) and differences in densities (i.e. shell walls may not be completely 'filled' with CaCO$_3$) may also determine the SNW. "*

**Response:** We agree and have added a section to the introduction stating this. However, preliminary results from ongoing work in our lab investigating the controls on porosity, and how it changes during ontogeny (by co-author Janet Burke) suggest that if anything changing porosity during ontogeny should work in opposition to the observed increase in CI with size. Therefore we do not dwell on this too much here.

**Comment:** *"page 3, line30: shouldn't this be '$\mu$ mol m$^{-2}$ s$^{-1}$'?"*
**Response:** Thank you for spotting this mistake. This has been corrected.

**Comment:** *"page 3, line 32: 'petri' should be capitalized."*
**Response:** Done

**Comment:** *"page 4, line 17: include 'mol/mol' after '6.25'."*
**Response:** Done

**Comment:** *"page 5, line 10: the 'area' reflects the size of the newly formed chambers. It should be noted here that the CI is therefore a species-specific measure (as is done in the appendix). When comparing CIs between species, morphology of the chambers should be taken into account (particularly when comparing spherical to more flattened chambers)."*
**Response:** Thank you for the suggestion. We have added a note to this effect at the end of this section.

**Comment:** *"page 5, line 24: Reiss (1957. Contribution from the Cushman Foundation for Foraminiferal Research 8, 127-145) is probably the first to have described the added layer of calcite on top of previously formed chambers. Moreover, this only applies to rotallid foraminifera (line 22), rather than foraminifera in general."*

**Response:** Thank you for highlighting Reiss's papers on this topic- we have cited him here. We have also clarified here that we are talking about planktonic (and so by implication rotalliid) foraminifera.

**Comment:** *"page 5, line 25: 'thickness' applies to the chamber wall, not to the chamber as such."*
**Response:** We have rephrased this sentence.

**Comment:** *"page 5, lines 27-28: does the thickening of the chamber walls in this case result in the formation of a crust?"*
**Response:** We distinguish 'crusting' from 'gametogenic' calcite, as others have (e.g. Hamilton et al., 2008, Marine Mi-
cropalaeontology). Hence we mention thickening during ontogeny in the following sentence rather than here. But we now use the word 'crust' to make this clearer.

**Comment:** *"page 7, line 7: 'more times' should be 'with one more layer' for this particular example."*
**Response:** Changed.

**Comment:** *"page 8, lines 32-33: sounds like a contradictory to me."*
**Response:** A 'not' was missing from the sentence here. Thank you!

**Comment:** *"page 8, line 34: effect of temperature on CI?"*
**Response:** Changed.

**Comment:** *"page 9, line 6: this is an important point, that may be stressed more in the discussion. A recently proposed model for (rotalid) foraminiferal calcification suggests that foraminifera are able to convert any DIC into $[CO_3^{2-}]$ at the site of calcification (Toyofuku et al., 2017. Nature Communications 8: 14145). This would hint to a primary control of [DIC] rather than ambient $[CO_3^{2-}]$, saturation state or seawater pH."*
**Response:** We now note the paper by Toyofuku et al. in the discussion (see below). However, our data do not show a significant relationship between residual CI and DIC ($R^2$ = -0.07, p = 0.553). We have noted this briefly here.

**Comment:** *"page 9, lines 25-26. This is not true. There are a number of papers providing an explanation for the contrasting*
*responses of marine calcifiers to ocean acidification by Justin Ries. Foremost, these include Ries et al., 2009 (Geology 37: 1131-1134) and Ries, 2011 (GCA 75: 4053-4064). Please include this in the discussion."*
**Response:** We now note these papers.

**Comment:** *"page 10, lines 17-32. I am not sure that I follow this completely. The distinction between smaller specimens that do not have an internal pool and larger foraminifera that require storage of DIC prior to chamber formation does make little sense to me. The references for small and larger specimens (Nehrke et al., 2013 versus Erez/ Ter Kuile) coincides with large inter-species differences (in morphology as well as carbonate chemistry). Without implying to impose my own favoured model for calcification (see e.g. Toyofuku et al., 2017), the basic mechanism in which protons are exchanged for calcium during*
*chamber formation is suggested for (benthic) species across a large range of sizes (Glas et al., 2012. Journal of Experimental Marine Biology and Ecology 424-425: 53-58). It may be, however speculative, that the steepness of the resulting external pH gradient (Glas et al., 2012; Toyofuku et al., 2017) determines the uptake rate of DIC. This steepness may well be a function of size and hence relates CI to volume/surface ratios. Depending on the authors' preference, I suggest to re-word this paragraph*

*somewhat. Without making a distinction between 'pools' versus 'no-pools', the mechanism leading to the uptake of DIC from*
*the surrounding seawater likely depends on the amount of ions necessary for the production of one new chamber (which is a function of the specimen's volume) and the surface area across which these ions have to be transported. These two units change with foraminiferal life-time and may hence result in a more or less cost-effective uptake of DIC (and $Ca^{2+}$).*"

**Response:** Thank you for your suggestion. We agree that the model proposed by Toyofuku et al. [2017] may also be consistent with our findings, and have extended this paragraph to explain that this is the case. However, because internal carbon
pools are unambiguously a feature of some foraminifera [e.g. ter Kuile et al. 1989], we think it may be informative to retain the existing discussion, although we also clearly state that the two models are not mutually exclusive.

**Comment:** *"page 11, line 10: it may be informative to include Flako-Zaritsky et al. (2011. Marine Micropaleontology 80:*
*74-88), who show that a number of benthic species are capable of producing their calcite despite very low pH and consequently, undersaturated conditions."*
**Response:** This reference and an accompanying explanatory sentence have been added.

**Comment:** *"page 13, line 11: there is a comma too much."*
**Response:** This sentence has been split and reworded for clarity.

**Comment:** *"figure 1: include panels (d) and (g) to the description for panel (a). Similar for (e)/(h) and (f)/(i)"*
**Response:** In order to avoid referencing panels in an inconsistent order, we now briefly describe panels d-i at the end of the caption.

**Comment:** *"figure 3: what happened to the uncertainties in the seawater Mg/Ca (panel a)?"*
**Response:** For Mg/Ca$_{sw}$, analytical uncertainty was $\pm 3\%$ [see Evans et al., 2015a] and as such is smaller than the symbol size. We have added a note explaining this in the figure caption.

**Comment:** *"figure 5: related to re-phrasing of the third paragraph of section 4.1, the authors may have to change the text*
*within this figure."*
**Response:** Text has been changed as requested to reflect the above suggestion.

**Comment:** *"table 3: CIs should have a unit ($\mu g/mm^2$)."*
**Response:** These have been added.

**Comment:** *"page 30, line 6: 'our' "*
**Response:** Replaced with 'the calcification'.

**Comment:** *"Figure A2: I don't understand how in model 2, b can vary by 10%, but never exceeds beyond $\sim 0.1075$ or drops below $\sim 0.0975$ instead of 0.110 and 0.900."*
**Response:** Many thanks for noticing this error. The figure has been updated accordingly. Note that whilst the wrong version
of this panel was included, there was no mistake in the underlying data analysis and as such it does not alter our conclusions in any way.